# Spatiotemporal control of photochromic upconversion through interfacial energy transfer

Long Yan[1], Jinshu Huang[1], Zhengce An[1], Qinyuan Zhang[1] & Bo Zhou [1] ✉

Dynamic control of multi-photon upconversion with rich and tunable emission colors is stimulating extensive interest in both fundamental research and frontier applications of lanthanide based materials. However, manipulating photochromic upconversion towards color-switchable emissions of a single lanthanide emitter is still challenging. Here, we report a conceptual model to realize the spatiotemporal control of upconversion dynamics and photochromic evolution of $Er^{3+}$ through interfacial energy transfer (IET) in a core-shell nanostructure. The design of Yb sublattice sensitization interlayer, instead of regular $Yb^{3+}$ doping, is able to raise the absorption capability of excitation energy and enhance the upconversion. We find that a nanoscale spatial manipulation of interfacial interactions between Er and Yb sublattices can further contribute to upconversion. Moreover, the red/green color-switchable upconversion of $Er^{3+}$ is achieved through using the temporal modulation ways of non-steady-state excitation and time-gating technique. Our results allow for versatile designs and dynamic management of emission colors from luminescent materials and provide more chances for their frontier photonic applications such as optical anti-counterfeiting and speed monitoring.

The development of smart luminescent materials with rich and switchable emission colors is important because of their diversities of frontier applications such as optical storage[1,2], multiple anti-counterfeiting[3–7], volumetric display[8–10], complex data analysis[11,12], monitoring[13,14], upconversion laser[15], and bioimaging and therapy[16–18]. Lanthanide-doped nanoparticles have multiple emission profiles covering ultraviolet, visible, and near-infrared spectral regions and become a class of ideal candidate for upconversion luminescence, showing unique characteristics of sharp emission peaks, abundant and tunable colors, long lifetime, flicker-free, photochemical stability and low toxicity[19–24]. It has been found that external stimuli approaches work well in tuning emission colors of lanthanides by altering excitation conditions[25–30] and applying pressure[31–33], temperature[34,35] and electric/magnetic fields[7,36,37]. In particular, the use of excitation

wavelengths and modes becomes a facile strategy to tune and switch the emission colors of lanthanide-doped nanoparticles with specific compositions and nanostructures[38–43]. Despite rapid progress in this field, the cooperative spatial and temporal control of upconversion dynamics for multiplexing photochromic evolution of a single lanthanide ion still remains a challenge.

Recently, some lanthanide lattices such as $NaErF_4$ show efficient upconversion and can be further used to tune emission colors by designing the excitation-emission orthogonal upconversion[44–48]. These materials can overcome the concentration quenching effect which usually exists in conventional luminescent materials[49]. In general, surface passivation is required to block energy loss to surface quenchers as a result of strong energy migration over the luminescent lattice. Considering the rich energy levels, it would be highly desirable

[1]State Key Laboratory of Luminescent Materials and Devices, Guangdong Provincial Key Laboratory of Fiber Laser Materials and Applied Techniques, and Guangdong Engineering Technology Research Center of Special Optical Fiber Materials and Devices, South China University of Technology, Guangzhou 510641, China. ✉e-mail: zhoubo@scut.edu.cn

to reach the color-switchable upconversion of a single lanthanide ion, instead of conventional two or more species of lanthanides, through rational manipulation of energy transfer channels. However, for these lanthanide lattices, it is theoretically and technically non-available to further enhance their emission intensity by increasing the emitter concentration (Fig. 1a, b). Enhancing the emitting brightness of lanthanide lattice becomes a bottleneck for their practical applications. The search for advanced strategies to break the concentration quenching effect plays a key role in the fundamental research of lanthanide luminescence physics and their frontier applications.

Here, we propose a conceptual model to realize the dynamic control of upconversion towards luminescence enhancement and switchable emission colors through interfacial energy transfer (Fig. 1c, d). The model is based on a core-shell-shell nanostructure consisting of a luminescent $NaErF_4$:Ho(0.5 mol%) core, a $NaYbF_4$ sensitization interlayer and an inert $NaYF_4$ protective shell layer (Fig. 1d). A small amount of $Ho^{3+}$ in $NaErF_4$ lattice helps for the redder emission of $Er^{3+}$ by promoting its populations from $^4I_{11/2}$ to $^4I_{13/2}$ through energy transfer looping from $Er^{3+}$ ($^4I_{11/2}$) to $Ho^{3+}$ ($^5I_6$) and then back to $Er^{3+}$ ($^4I_{13/2}$), and other energy transfers between $Er^{3+}$ and $Ho^{3+}$ can be ignored (Supplementary Figs. 1, 2)[44]. The harvest capability of 980 nm excitation light can be improved due to the presence of $NaYbF_4$ sensitization layer, which further helps to enhance the upconversion emissions of $Er^{3+}$ by the $Yb^{3+}$-to-$Er^{3+}$ IET channel (Fig. 1c). More importantly, this model is able to realize the spatiotemporal control of upconverted emission colors from a single $Er^{3+}$ ion by finely and precisely manipulating interfacial interactions between Er and Yb sublattices. We found that it is easy to tune emission colors by altering pump laser power, laser pulse width and temporal observation window due to the different rise and decay times. Such a conceptual design not only presents a perspective to manipulation of ionic interactions between heavily doped lanthanides, but also works well in generating color-switchable emissions through multi-dimensional modulation, which is believed to be an ideal candidate for smart materials with great potential in optical information security, anti-counterfeiting and velocity measurement.

## Results
### Enhancing upconversion through IET channel

We first synthesized a series of $NaErF_4$:Ho(0.5 mol%) @$NaYF_4$:Yb(0−100 mol%)@$NaYF_4$ core-shell-shell nanoparticles by a modified coprecipitation and thermal decomposition method. The samples show uniform sphere morphology and monodisperse distribution (Fig. 2a left panel; Supplementary Fig. 3). A lattice fringe distance of 0.51 nm is clearly observed from the high-resolution TEM image (Fig. 2a right top panel), corresponding to the *d*-spacing of (100) lattice planes of the sample. The samples are in hexagonal phase according to the Fourier transform diffraction pattern and the X-ray diffraction data (Fig. 2a, right bottom panel; Supplementary Fig. 4). The core-shell-shell nanostructure is also confirmed by the element mapping result showing a contrast in the distributions of different lanthanide species (Fig. 2b).

Figure 2c shows the upconversion emission spectra of the samples under 980 nm excitation. Typical upconverted green and red emissions of $Er^{3+}$ were observed, corresponding to its ($^2H_{11/2}$,$^4S_{3/2}$) → $^4I_{15/2}$ and $^4F_{9/2}$ → $^4I_{15/2}$ transitions, respectively. With the increase of $Yb^{3+}$ concentration in the sensitization interlayer, the upconversion emission shows a significant monotonous enhancement, and the red emission is always much greater than the green emission, as a result of cross relaxations between $Er^{3+}$ ions such as ([($^2H_{11/2}$,$^4S_{3/2}$); $^4I_{9/2}$)] → [$^4F_{9/2}$; $^4F_{9/2}$][44,47]. As a control, other excitation wavelengths (e.g., 808 and 1530 nm) do not lead to an enhancement of upconversion because they have no response to the absorption of $Yb^{3+}$ (Fig. 2d). Instead, a heavy luminescence quenching was obversed at high $Yb^{3+}$

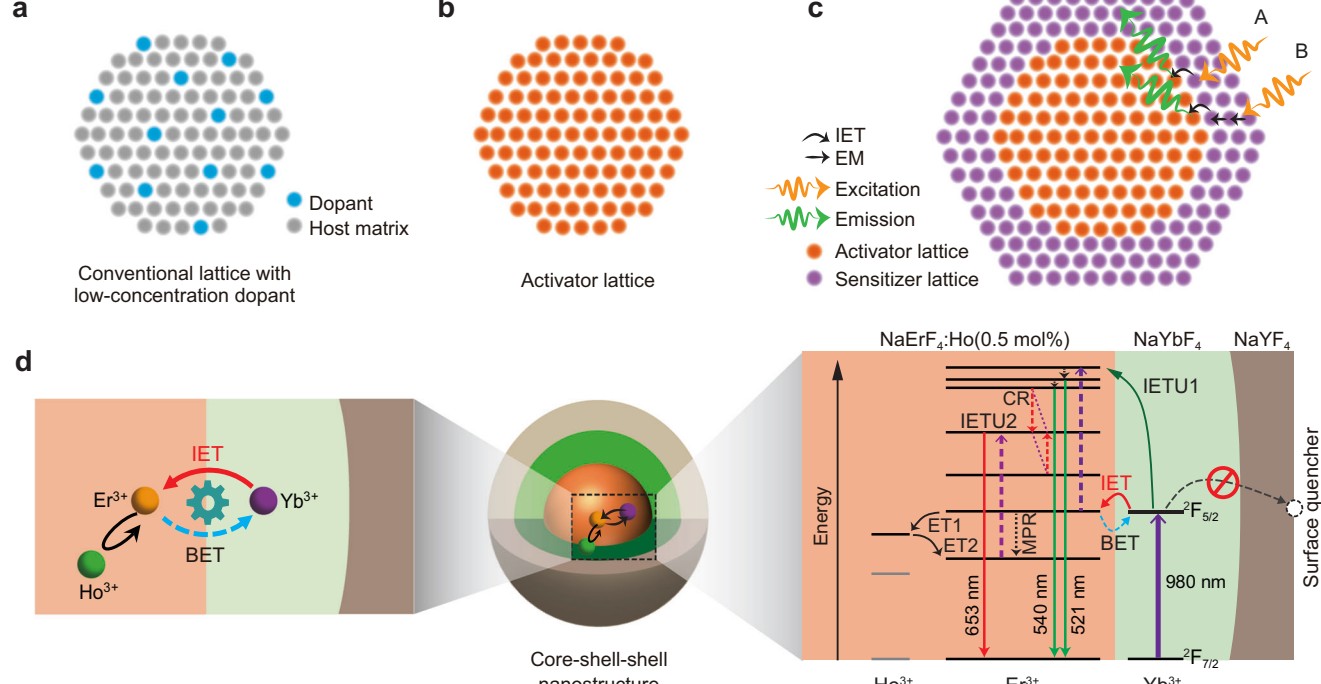

**Fig. 1 | Schematic of tuning upconversion through interfacial energy transfer in a core-shell-shell nanostructure. a, b** Comparison of conventional lattice with low-concentration dopant and the lanthanide lattice for upconversion. **c** Schematic of using IET to control upconversion of lanthanide lattice by constructing active core-sensitizing shell structure. The excitation A presents the direct IET from $Yb^{3+}$ to $Er^{3+}$, and excitation B presents the IET with the help of energy migration (EM) among the ytterbium lattice. **d** Proposed conceptual model and possible energy transfer channels of $NaErF_4$:Ho(0.5 mol%)@$NaYbF_4$@$NaYF_4$ core-shell-shell nanostructure to enhance upconversion and tune its emission colors through spatiotemporal control of IET between erbium and ytterbium lattices. ET, BET, IETU and MPR stand for energy transfer, back energy transfer, interfacial energy transfer upconversion and multi-phonon relaxation, respectively.

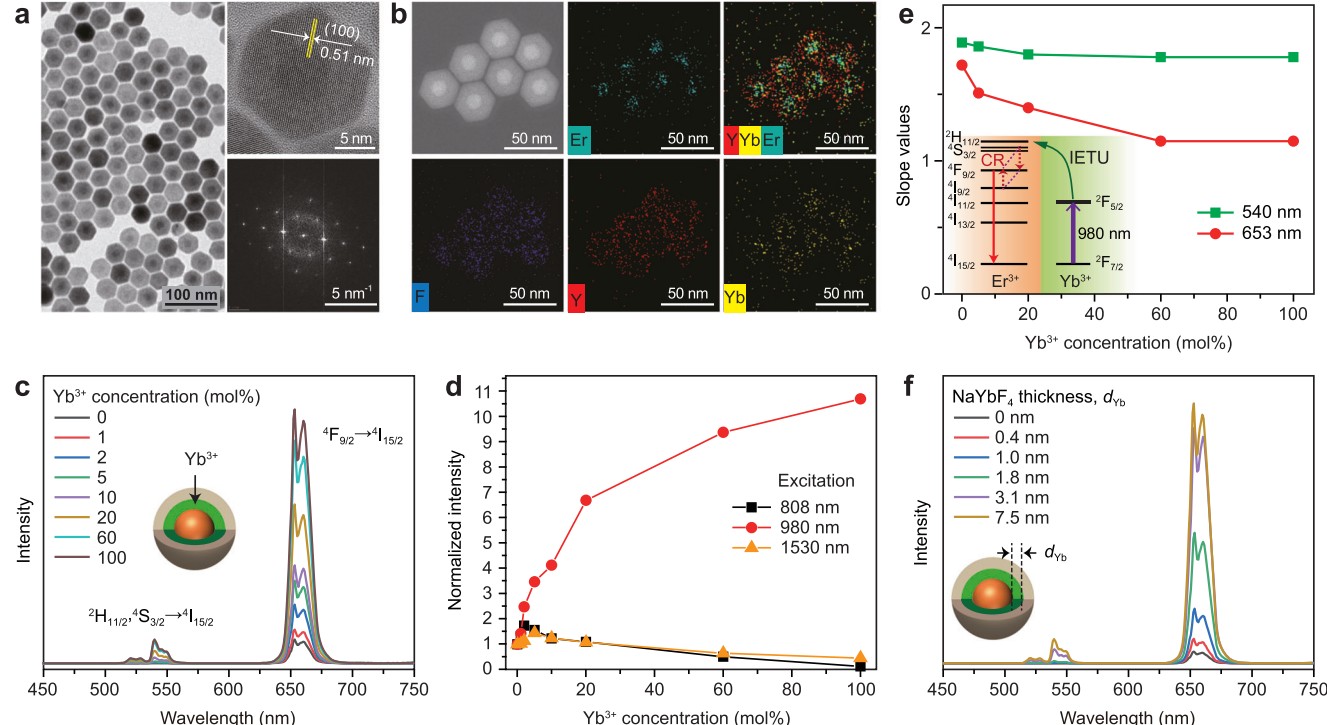

**Fig. 2 | Characterization and upconversion emission spectra. a** TEM image, high resolution TEM image (right top panel) and the corresponding Fourier transform diffraction pattern (right bottom panel) of the as-synthesized NaErF₄:Ho(0.5 mol%)@NaYF₄:Yb(20 mol%)@NaYF₄ core-shell-shell nanoparticles. **b** STEM image of (**a**) sample and corresponding elemental mappings of Er, Yb, Y and F. **c** Upconversion emission spectra of NaErF₄:Ho(0.5 mol%)@NaYF₄:Yb(0–100 mol%)@NaYF₄ core-shell-shell nanoparticles with a fine tuning of Yb³⁺ dopant concentration in the sensitizing layer under 980 nm excitation. **d** Dependence of normalized upconversion emission intensity on Yb³⁺ dopant concentration of (**c**) sample under 808,

980 and 1530 nm excitations, respectively. The data were normalized to the red emission intensity of (**c**) sample without Yb³⁺ doping. **e** Dependence of slope values of upconversion emissions on Yb³⁺ concentration for (**c**) samples under 980 nm excitation. Inset shows possible cross relaxations to populate the ⁴F₉/₂ state of Er³⁺. **f** Upconversion emission spectra of NaErF₄:Ho(0.5 mol%)@NaYbF₄@NaYF₄ core-shell-shell nanoparticles with a fine tuning of NaYbF₄ layer thickness (0–7.5 nm) under 980 nm excitation. All excitation power densities were 11.6 W cm⁻². Source data are provided as a Source Data file.

concentrations owing to the Er³⁺-Yb³⁺ energy transfer (Supplementary Fig. 5). More importantly, the upconversion quantum yield was also increased by comparison to the control sample without the NaYbF₄ interlayer (Supplementary Fig. 6). In addition, ion diffusion at the core-shell interfacial region can be ignored (Supplementary Figs. 7, 8)[50-53]. These observations confirm that our IET design works well in enhancing the upconversion of Er sublattice.

The pump power dependent upconversion was also investigated (Supplementary Fig. 9). An interesting observation is that the slope value of red emission exhibits an obvious decline as Yb³⁺ concentration increases (Fig. 2e). The presence of Yb³⁺ in the interlayer can further improve the population of Er³⁺ at its green emitting levels through IET and promote the cross relaxation of [(²H₁₁/₂,⁴S₃/₂); ⁴I₉/₂)] → [⁴F₉/₂; ⁴F₉/₂]. This adds an additional channel to increase the population of red emitting ⁴F₉/₂ level and reduces the slope value. Such difference in slope values of red and green emissions presents a facile way to tune emission colors by elevating pump power densities (Supplementary Fig. 10).

It is further found that thicker NaYbF₄ interlayer helps to improve the upconversion luminescence of the samples (Fig. 2f; Supplementary Figs. 11, 12). For instance, the red upconversion emission was enhanced over 24.5 folds than that of the NaErF₄:Ho@NaYF₄ control sample under identical measurement condition (Supplementary Fig. 13). These results suggest that increasing sensitizer content via thickening the sensitization layer is a good strategy to raise the energy harvest capability of the incident 980 nm irradiation and promote the IET at the core-shell interface. This is also supported by the increased absorption spectra of the samples (Supplementary Fig. 14). However, a

saturation trend of the upconversion intensity was observed when the thickness is over 3.1 nm, might be due to the energy migration over Yb³⁺ sublattice which would reduce the energy transfer to activators (Supplementary Fig. 15).

## Spatial control of upconversion

To further understand the IET-induced upconversion luminescence enhancement, we investigated the possible energy transfer channels that may occur at the core-shell interfacial region. The dependence of the upconversion emissions on Yb³⁺ content in the sensitization interlayer were compared in Fig. 3a, showing a rapid increase for both red and green emissions when Yb³⁺ concentration lies in 0-20 mol% range. In contrast, the absorption spectrum at 980 nm shows a continuous increase (Fig. 3a, b), suggesting that the presence of Yb³⁺ in the sensitization interlayer indeed promotes the absorbance of 980 nm laser energy markedly. Another merit is the much higher upconversion enhancement factor at lower pump power densities (Supplementary Fig. 16), which is important for bioapplications. Considering the energy level diagram of Er³⁺ and Yb³⁺, back energy transfer may occur for the green and red emitting energy levels of Er³⁺ (Supplementary Fig. 17a). To shed more light on this issue, we measured the decay curves of the upconversion emissions (Supplementary Fig. 17b–d), and found that the lifetime values of both green and red emissions present a decline when Yb³⁺ concentration is over 5 mol%. This implies that there exists a competition between the forward and backward energy transfer channels between Er³⁺ and Yb³⁺ at the core-shell interface for the samples with high Yb³⁺ content, resulting in a saturation trend in upconversion intensity.

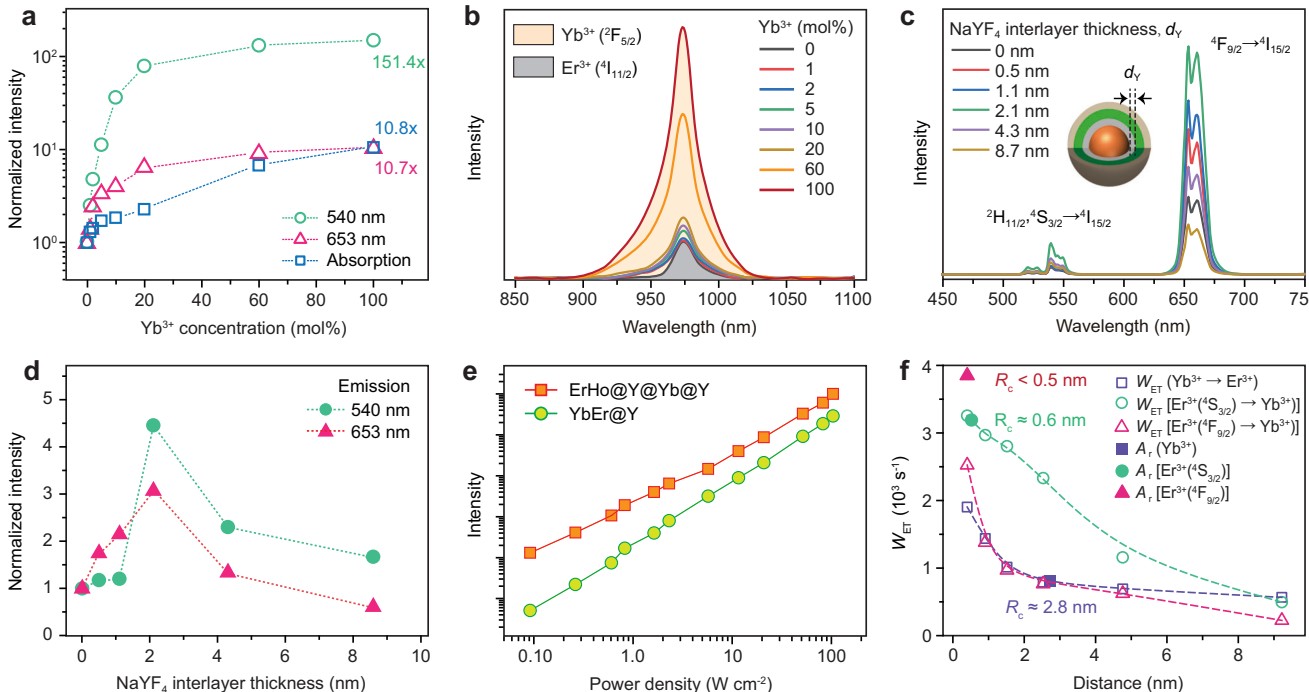

**Fig. 3 | Spatial control of upconversion. a** Normalized upconversion emission intensity of NaErF$_4$:Ho(0.5 mol%)@NaYF$_4$:Yb(0–100 mol%)@NaYF$_4$ core-shell-shell nanoparticles with variable Yb$^{3+}$ concentrations. The absorption intensity was also plotted for comparison. The data were normalized to the sample with 0 mol% Yb$^{3+}$ content. **b** Absorption spectra of (**a**) samples due to Yb$^{3+}$ ($^2F_{5/2} \leftarrow ^2F_{7/2}$ transition) and Er$^{3+}$ ($^4I_{11/2} \leftarrow ^4I_{15/2}$ transition). **c** Upconversion emission spectra of NaErF$_4$:Ho(0.5 mol%)@NaYF$_4$@NaYbF$_4$@NaYF$_4$ samples with a fine tuning of the NaYF$_4$ interlayer thickness from 0 to 8.7 nm. **d** A comparison of emission intensity

of **c** samples normalized to that without the NaYF$_4$ interlayer. **e** A comparison of upconversion emission intensity from (**c**) sample with 2.1 nm thick NaYbF$_4$ inter-layer (ErHo@Y@Yb@Y) and control NaYF$_4$:Yb/Er(20/2 mol)@NaYF$_4$ core-shell nanoparticles (YbEr@Y) under identical measurement condition. **f** Dependence of energy transfer rate between Yb$^{3+}$ and Er$^{3+}$ on the ionic distance. $W_{ET}$ and $A_r$ represent energy transfer rate and spontaneous emission rate, respectively. All excitation power densities were 11.6 W cm$^{-2}$. Source data are provided as a Source Data file.

To further enhance upconversion, we proposed a nanoscale control of Er$^{3+}$-Yb$^{3+}$ spatial separation by inserting a thin optically inert NaYF$_4$ interlayer between the core and sensitization layer. Namely, the NaErF$_4$:Ho@NaYF$_4$@NaYbF$_4$@NaYF$_4$ core-multishell nanoparticles with a fine tuning of NaYF$_4$ interlayer thickness from 0 to 8.7 nm were synthesized (Supplementary Figs. 18, 19). Their upconversion emissions show an initial increase before a decline with elevating the inert NaYF$_4$ interlayer thickness (Fig. 3c). When it reaches 2.1 nm, the upconversion emission intensity is enhanced over threefolds (Fig. 3d). This means that a suitable nanoscale spatial separation of Er and Yb sublattices can effectively reduce the energy loss due to the backward energy transfer channels. This is also supported by the prolonged lifetimes of red and green emissions (Supplementary Fig. 20) and increased quantum yield values (Supplementary Fig. 21). Here it is worth noting that the upconversion luminescence of this sample is much higher than that of the commonly reported NaYF$_4$:Yb/Er(20/2 mol%)@NaYF$_4$ core-shell nanoparticles, albeit a decrease in quantum yield (Fig. 3e; Supplementary Figs. 22, 23).

The quenching processes for the green and red emissions can be assigned to the back energy transfer from Er$^{3+}$ to Yb$^{3+}$ at the core-shell interfacial region, which show a sensitive dependence on ionic separation. We further calculated the critical distance ($R_c$) at which the energy transfer rate is equal to the emission rate for a donor ion (Fig. 3f; Supplementary Fig. 24)[49,54,55]. It is found that $R_c$ is no more than 0.6 nm for the backward energy transfer, much shorter than the forward energy transfer ($\approx$2.8 nm). Therefore, the nanoscale manipulation of the ionic interactions through IET is capable of suppressing the unwanted quenching channels and improving the utilization of excitation energy for efficient upconversion.

## Temporal control of upconversion

In general, energy transfer upconversion exhibits a typical rise time, providing a temporal means to control the upconversion dynamics in addition to the spatial manipulation. We first investigated the temporal feature of Er$^{3+}$ upconversion in the NaErF$_4$:Ho@NaYF$_4$:Yb(0–100 mol%)@NaYF$_4$ samples under pulsed 980 nm laser with variable pulse widths (Fig. 4a; Supplementary Figs. 25, 26). Interestingly, the green upconverted emission shows a close dependence on the pulse width of the excitation laser, and when Yb$^{3+}$ content locates in the range of 10–60 mol% the samples show clear green emission color (Fig. 4b, c). For example, the sample shows a gradual color change from red to yellow and then green as the pulse width was reduced from 10 to 0.5 ms (Fig. 4c). This suggests that a precise emission color tuning becomes easily available by modulating the pulse width of the 980 nm laser. Other samples with thickness-tunable NaYbF$_4$ sensitization interlayers and with Er-Yb spatial separations produce similar color changes (Supplementary Figs. 27–31). These results confirm that temporal control of upconversion dynamics is an effective and convenient way to manipulate the output of emission colors.

To further understand such an interesting phenomenon, we measured the time-dependent emission profiles. As shown in Fig. 4d and Supplementary Fig. 32, both green and red emissions show a rapid rise process as the Yb$^{3+}$ content increases in the sensitizing interlayer. More interestingly, the rise time of the green emission becomes faster than that of the red emission. This differs greatly from the NaErF$_4$@NaYF$_4$ and NaErF$_4$:Ho@NaYF$_4$ control samples, which show slower or comparable rise time for green upconversion with only red light output by reducing the pulse width of excitation laser (Supplementary Fig. 33). The faster rise time of the green

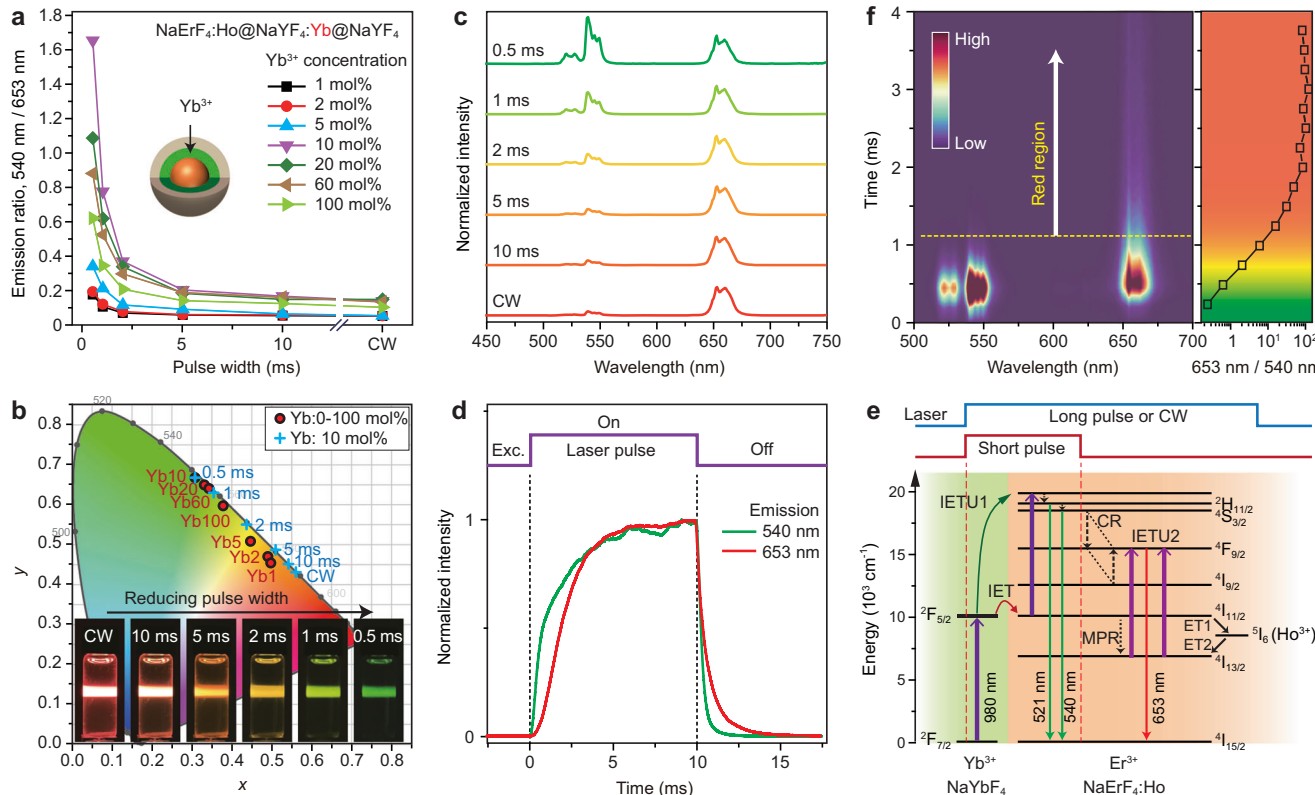

**Fig. 4 | Temporal control of upconversion. a** Dependence of green-to-red upconversion emission intensity ratio on the pulse width of 980 nm excitation for the NaErF$_4$:Ho(0.5 mol%)@NaYF$_4$:Yb(0–100 mol%)@NaYF$_4$ core-shell-shell nanoparticles. **b** CIE coordinates of the emission colors of (**a**) samples under 980 nm excitation with pulse width of 0.5 ms (red label) and that of NaErF$_4$:Ho(0.5 mol%)@NaYF$_4$:Yb(10 mol%)@NaYF$_4$ core-shell-shell nanoparticles under 980 nm excitation with different pulse widths (blue label). Insets show their emission photographs. **c** Normalized upconversion emission spectra of NaErF$_4$:Ho(0.5 mol%)

@NaYF$_4$:Yb(10 mol%)@NaYF$_4$ core-shell-shell nanoparticles under 980 nm excitation with different pulse widths. **d** Time-dependent emission profiles of Er$^{3+}$ measured at 540 and 653 nm for **c** sample. **e** Schematic of possible mechanism for the upconversion emission color change under non-steady-state excitation. **f** Two-dimensional time-resolved photoluminescence spectra of NaErF$_4$(0.5 mol%) @NaYF$_4$:Yb(20 mol%)@NaYF$_4$ core-shell-shell nanoparticles under 980 nm excitation. All excitation power densities were 11.6 W cm$^{-2}$. Source data are provided as a Source Data file.

emission might be resulted from the resonant energy transfer from Yb$^{3+}$ in the sensitization interlayer to Er$^{3+}$ in the core. While for the red emission, it needs an assistance of multi-phonon relaxation ($^4I_{11/2}$-to-$^4I_{13/2}$) or energy transfer looping from Er$^{3+}$ ($^4I_{11/2}$) to Ho$^{3+}$ ($^5I_6$) and then back to Er$^{3+}$ ($^4I_{13/2}$), which results in a delay in the population of the intermediate $^4I_{13/2}$ level and subsequent red-emitting $^4F_{7/2}$ level. After the initial rising time stage, some of the Er$^{3+}$ ions at the green emitting levels would experience a severe depopulation due to cross relaxation. In this case, the population of Er$^{3+}$ at the red emitting $^4F_{7/2}$ level can be improved with a decline of the intensity-power slope values (Fig. 2e).

Another interesting result is the sharp contrast in the decay times of upconverted emissions. As shown in the time-resolved luminescence spectra (Fig. 4f; Supplementary Fig. 34), the red emission has a much longer decay time than that of the green emission. In the initial stage after switching-off the excitation laser, the sample shows dominant green light due to its faster rise time than that of the red emission. While when the time is over 1 ms, the sample begins to show red light owing to its slower decay rate, resulting in a color change under different observation time. A tuning of excitation pulse widths imposed an impact on the decays, and it shows longer lifetime at wider pulse width may be owing to the possible energy migration among Yb$^{3+}$ ions (Supplementary Fig. 35)[27]. While it should be noted that the lifetime of red emission is always longer than that of green emission under different pulse width excitations. Other samples exhibit a similar lifetime feature (Supplementary Tables 1–4).

## Advanced optical applications

The smart control of emission colors in nanoparticles permits great chances for frontier photonic applications. As shown in Fig. 5a, the temporal tuning of photochromic evolution helps to improve the optical anti-counterfeiting performance. For instance, the hidden information of "SCUT" pattern in a QR code was identified by increasing the power of 980 nm CW excitation laser or utilizing short pulse laser irradiation, and clear information can be further extracted by a 600 nm short-pass filter (Fig. 5b; Supplementary Table 5). Such a multi-mode strategy to obtain the optical signal confirms the advantage of decoding the concealed information. In addition, the tunable emitting colors can also be implemented in fast recognition of multi-dimensional optical signals (Fig. 5c).

On the other hand, the temporal dependent emitting colors are usually difficult to observe directly for human eyes, which require complex and expensive instrument. Here, we proposed a simple turntable design to convert temporal photochromic evolution to spatial color distribution. As shown in Fig. 5d, a fixed CW 980 nm laser was used to irradiate the sample that rotates with the disk. In this case, the rotational speed could affect the excitation frequency and irradiation duration (equivalent to pulse width). In the section I region, the emission shows the color at the rising edge similar to that under pulse excitation. Improving rotational speed leads to an increment of the frequency and contraction of pulse width. In the section II region, the emission with longer lifetime began to be identified, and increasing the rotational speed can extend the length

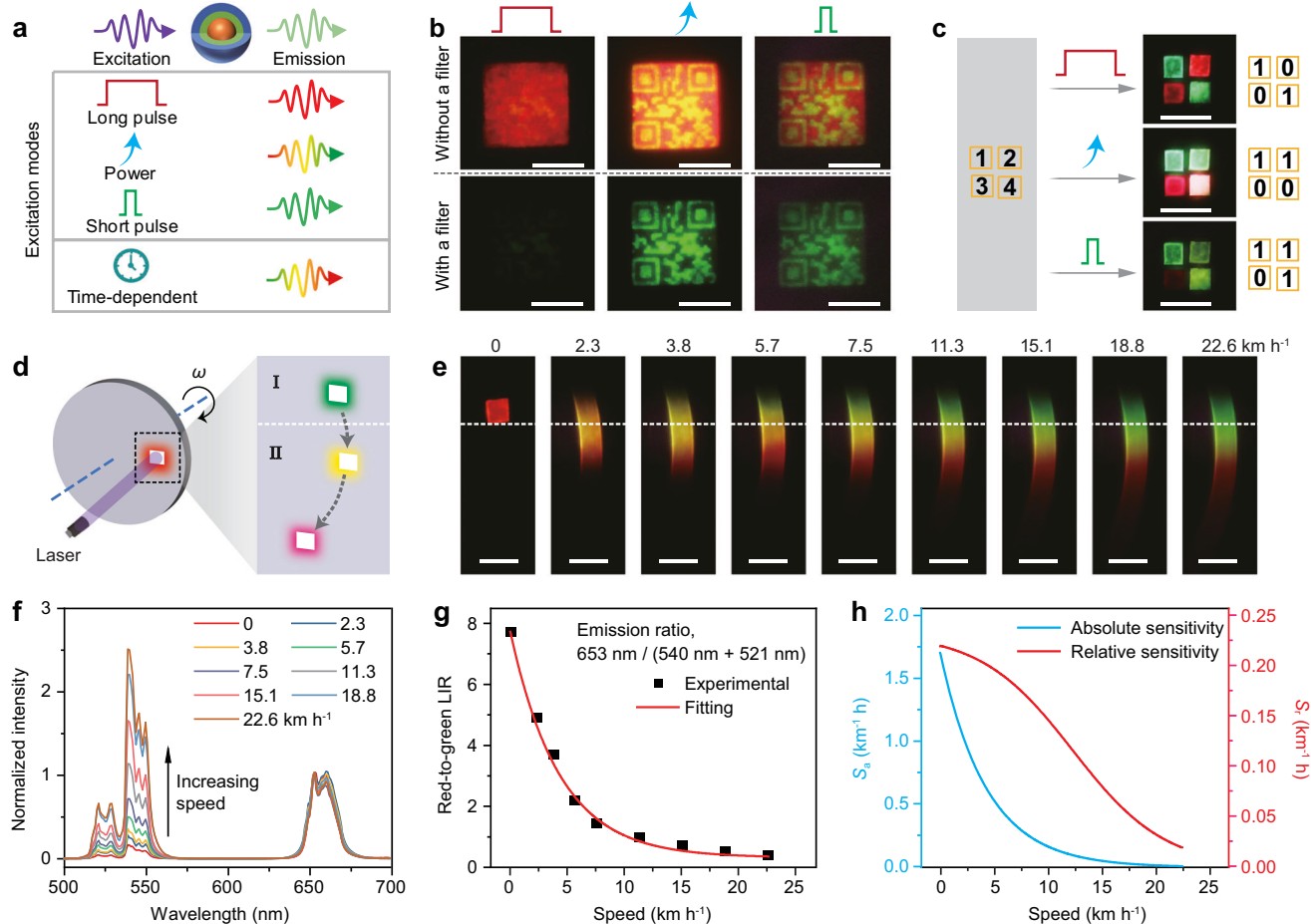

**Fig. 5 | Application of color-switchable upconversion. a** Schematic of multi-mode modulation of upconversion emission colors. Long pulse (width >10 ms) and short pulse (width <1 ms) excitations lead to red and green emission colors, respectively, and a tuning of excitation pump power or pulse width results in a gradual color change between red and green. **b** Decoding the pattern information (left panels) through high pump power (middle panels) or short pulse irradiation (right panels). The photographs were taken with and without a 600 nm short-pass filter, respectively. Scale bar, 5 mm. **c** Fast recognition of multidimensional optical signals. Scale bar, 5 mm. **d** Schematic of transforming luminescence information from temporal to spatial dimension. Scale bar, 5 mm. **e** Photographs of the rotating turnplate with increasing speed under steady-state 980 nm laser excitation. The corresponding speed (km/h) is marked at the bottom. **f** Speed-dependent upconversion emission spectra of NaErF$_4$:Ho(0.5 mol%)@NaYF$_4$:Yb(20 mol%)@NaYF$_4$ core-shell-shell nanoparticles under steady-state 980 nm excitation. The spectra were normalized to red emission intensity. **g** Plot of luminescence intensity ratio (LIR) versus speed together with the fitting by an exponential function. **h** Dependence of calculated absolute ($S_a$) and relative ($S_r$) sensitivity on speed. Source data are provided as a Source Data file.

of the emission bar, accompanied by an obvious trend of emitting color change to red (Fig. 5e, f; Supplementary Fig. 36). As a control, such interesting light color evolution was not achievable in conventional upconversion nanoparticles such as NaYF$_4$:Yb/Er(20/2 mol%)@NaYF$_4$ and NaErF$_4$@NaYF$_4$ (Supplementary Figs. 37, 38). More interestingly, the red-to-green luminescence intensity ratio exhibits a exponential dependence on speed (Fig. 5g; Supplementary Table 6), and the maximum speed sensitivity was further calculated to be 1.70 km$^{-1}$ h (Fig. 5h), higher than the recent reports[13,56], suggesting a good candidate for speed monitoring.

## Discussion

We have demonstrated that interfacial energy transfer is an effective strategy to spatiotemporally control the upconversion dynamics of erbium sublattice. The upconversion luminescence of Er sublattice was enhanced through IET from the sensitizing Yb sublattice interlayer, and the quantum yield was also increased. A fine spatial manipulation of forward and backward energy transfer further contributed to the upconversion luminescence. Moreover, the green/red dual-color switchable output was realized by precise control of the rise and decay dynamics through using short pulse laser excitation and time-

gating technique. By employing these optical functional materials, the decryption level was improved, and the transformation of optical information from temporal to spatial dimensions was also realized, showing great potential in the speed monitoring. Our results present an in-depth insight into the photophysics involved in the interfacial interactions between lanthanide lattices instead of conventional lanthanide doping. More significantly, they would contribute to the development of smart luminescent materials towards frontier photonic applications.

## Methods
### Materials

The lanthanide raw materials including erbium(III) acetate hydrate (99.9%), yttrium(III) acetate hydrate (99.9%), ytterbium(III) acetate hydrate (99.99%), holmium(III) acetate hydrate (99.9%), yttrium oxide (99.9%), ytterbium oxide (99.9%), yttrium(III) oxide (99.9%), ytterbium(III) oxide (99.9%) together with oleic acid (90%), 1-octadecene (90%), sodium hydroxide (NaOH; >98%), ammonium fluoride (NH$_4$F; >98%), and trifluoroacetic acid (CF$_3$COOH; >99%) were purchased from Sigma-Aldrich. The sodium trifluoroacetate (CF$_3$COONa; >98%) were purchased from Alfa. All of these materials were used as received unless otherwise noted.

## Synthesis of core nanoparticles

The core nanoparticles were synthesized using a modified co-precipitation chemical method. In a typical procedure for the synthesis of $NaErF_4$:(Ho 0.5 mol%) core nanoparticle, to a 250-mL flask containing oleic acid (33 mL) and 1-octadecene (67 mL) was added a water solution containing 3.98 mmol $Er(CH_3CO_2)_3$ and 0.02 mmol $Ho(CH_3CO_2)_3$. The resulting mixture was heated at 150 °C for 1 h and then cooled down to room temperature. Subsequently, a methanol solution containing NaOH (10 mmol) and $NH_4F$ (16 mmol) was added and stirred at 50 °C for 1 h, and then heated to 100 °C under vacuum for 0.5 h. After passing argon flow, the system was quickly heated to 300 °C and kept for 1.5 h before cooling down to room temperature. The resulting core nanoparticles were collected by centrifugation (7000 × g, 5 min), washed with ethanol, and finally dispersed in 40 mL cyclohexane. Other core nanoparticles except for $NaYbF_4$:Er(1 mol%) were synthesized using a similar procedure except for different designed ratios of lanthanide precursors.

The $NaYbF_4$:Er(1 mol%) nanoparticles were synthesized using a high temperature injection method. In a typical procedure for the core precursor, to a 50-mL flask containing oleic acid (2 mL) and 1-octadecene (5 mL) was added a water solution containing 0.396 mmol $Yb(CH_3CO_2)_3$ and 0.004 mmol $Er(CH_3CO_2)_3$. The resulting mixture was heated at 150 °C for 1 h and then cooled down to room temperature. Subsequently, a methanol solution containing NaOH (1 mmol) and $NH_4F$ (1.6 mmol) was added and stirred at 45 °C for 2 h, then heated to 100 °C under vacuum for 10 min and cooled to room temperature. A mixture of OM (5 mL) and ODE (5 mL) as substrate was loaded in a 50-mL round bottom flask and heated to 120 °C. It was degassed through a vacuum pump for 20 min before the temperature was raised to 300 °C under argon protection. After the temperature of substrate reached 300 °C, the core precursor was quickly injected in one-shot, and the reaction media were stirred at 300 °C for 60 min. After the reaction was cooled down to room temperature, the resulting core nanoparticles were collected by centrifugation (7000 × g, 5 min), washed with ethanol and methanol, and finally dispersed in 4 mL cyclohexane.

## Synthesis of core-shell, core-shell-shell and core-multishell nanoparticles

The core-shell nanoparticles were synthesized using a modified thermal decomposition method. The 2 mL as-prepared $NaErF_4$:Ho(0.5 mol%) core nanoparticle were used as seeds. The oleic acid (10 mL), 1-octadecene (10 mL), $Na(CF_3CO_2)_3$ (0.8 mmol), aqueous solution containing $Yb(CF_3CO_2)_3$ (0.8 mmol) were added to a 100-mL flask (with core/shell mass ratio of 1:4). The mixture was heated and stirred at 120 °C for 40 min under argon flow and then cooled down to room temperature. Cyclohexane (2 mL) containing core nanoparticle was added in precursor, and then the solution was degassed at 120 °C for 30 min to remove residual moisture and cyclohexane. Afterwards, the system was switched to an argon flow and quickly heated to 300 °C for 30 min. The resulting $NaErF_4$:Ho(0.5 mol%)@$NaYbF_4$ core-shell nanoparticles were collected by centrifugation (7000 × g, 5 min), washed with ethanol, and finally dispersed in 2 mL cyclohexane. The other core-shell, core-shell-shell and core-multishell samples were synthesized by a similar process except for the different fractions of precursors.

The core-shell based nanoparticles with different shell thickness were synthesized by the same process, and the shell thickness was controlled by tuning the amount of reactant substance. Taking the $NaErF_4$:Ho(0.5 mol%)@$NaYbF_4$ core-shell nanoparticles with thickness-tunable $NaYbF_4$ for example, the core/shell molar ratios of 1:0.4, 1:0.7, 1:1, 1:2, 1:4 were used, namely, the shell precursors included 0.08, 0.14, 0.2, 0.4, 0.8 mmol $Yb(CF_3CO_2)_3$ and an equal amount of $Na(CF_3CO_2)_3$. Other shells in a core-multishell nanostructure with tunable thicknesses was synthesized by a similar process except for using the different precursors.

## Characterization

Powder X-ray diffraction (XRD) and in-situ XRD patterns of selected samples at elevated temperatures were detected on an Aeris powder XRD diffractometer (PANalytical Corporation, Netherlands) operating at 40 kV and 15 mA with monochromatized Cu Kα radiation ($\lambda$ = 1.5406 Å). The low- and high-resolution transmission electron microscopy (TEM) measurements together with energy-dispersive X-ray spectroscopy (EDS) were carried out on a JEOL JEM-2100F transmission electron microscope with an acceleration voltage of 200 kV. The upconversion emission spectra were detected by a Zolix spectrofluorometer equipped with external power-controllable laser diodes of 980, 808 and 1530 nm. The pulse width dependent spectra and decay curves were detected by the same spectrophotometer equipped with a Tektronix oscilloscope (MDO32).

## Upconversion quantum yield measurement

The upconversion quantum yield (QY) was measured by an integrating sphere connected to a fiber optic spectrometer (see Supplementary Fig. 39). The setup was first calibrated by using the standard lamp (Tungsten-Halogen Lamp; Gilway 187-1) to obtain the correction factor, which was used to correct the measured spectra of the sample. Next, a side-polished quartz cuvette containing a cyclohexane solution of the nanoparticles was placed inside the integrating sphere coupled with the fiber optic spectrometer (QE65 Pro, Ocean optics), which collects the emission and excitation light signals under 980 nm excitation. A cyclohexane solution without the nanoparticles was also used for reference. The QY can be calculated by the following equation:

$$QY = N_{em}/N_{abs} = \int P_{em}(\lambda)\lambda \mathrm{d}\lambda / \int \left[ P_{ref}(\lambda) - P_{sample}(\lambda) \right] \lambda \mathrm{d}\lambda \qquad (1)$$

where $P_{em}(\lambda)$ is the upconversion emission power spectrum of the sample, $P_{ref}(\lambda)$ and $P_{sample}(\lambda)$ are the power spectra of excitation light recorded after passing through the reference and sample, respectively.

## Energy transfer rate calculation

The donor-acceptor energy transfer rate ($W_{ET}$) can be calculated by

$$W_{ET} = \frac{1}{\tau} - \frac{1}{\tau_0} \qquad (2)$$

where $\tau$ and $\tau_0$ are the lifetime of a given donor ion with and without codoping an acceptor ion, respectively.

## Luminescent pattern of QD code

The QD code and square pattern were printed by screen printing method. The QD code was printed using $NaErF_4$:(Ho 0.5 mol%)@$NaYF_4$:Yb(20 mol%)@$NaYF_4$ core-shell-shell nanoparticles and the rest of square pattern was printed by $NaErF_4$:Ho(0.5 mol%)@$NaYF_4$:Yb(10 mol%) core-shell nanoparticles. Under different excitation conditions, the QR code could be decoded due to the color change from red to green. And only the green emission regions could be identified by using a 600 nm short-pass filter under same condition.

## Fast recognition of multidimensional optical signals

The pattern consists of four small square patterns printed by $NaYF_4$:Yb/Er(20/2 mol%)@$NaYF_4$ core-shell, $NaErF_4$:Ho(0.5 mol%)@$NaYF_4$:Yb(20 mol%)@$NaYF_4$ core-shell-shell, $NaErF_4$:Ho(0.5 mol%)@$NaYF_4$ and $NaYbF_4$:Er(1 mol%)@$NaYF_4$ core-shell nanoparticles, respectively. Under suitable excitation conditions, the emission color of different regions could be recognized rapidly. The sample for emission color tuning under different excitation conditions was summarized in Supplementary Table 5.

## Control of photochromic upconversion and speed sensitivity measurement

The $NaErF_4$:Ho(0.5 mol%)@$NaYF_4$:Yb(20 mol%)@$NaYF_4$ nanoparticles were sprayed into 3 mm × 3 mm square pattern on the rotary table. The linear speed was determined by the revolution speed of the stepper motor and radial distance of samples. The speed rate was measured by the luminescence intensity ratio (LIR) technique. The experimental LIR data can be fitted by an exponential function, and the absolute sensitivity ($S_a$) and relative sensitivity ($S_r$) can be calculated by the following equations:

$$S_a = \frac{dLIR}{d\nu} \tag{3}$$

$$S_r = \frac{1}{LIR}\frac{dLIR}{d\nu} \tag{4}$$

where $\nu$ is linear speed.

## Reporting summary

Further information on research design is available in the Nature Portfolio Reporting Summary linked to this article.

## Data availability

The data that support the findings of this study are available from the corresponding author upon reasonable request. Source data are provided with this paper.

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

## Acknowledgements

This work was supported by the National Natural Science Foundation of China (Grant Nos 52272151 and 51972119), the Fundamental Research Funds for the Central Universities (2022ZYGXZR015 and 2023ZYGXZR053), the China Postdoctoral Science Foundation (2022M721178), and the Local Innovative and Research Teams Project of Guangdong Pearl River Talents Program (2017BT01X137).

## Author contributions

B.Z. conceived the concept. B.Z. and Q.Z. supervised the project. L.Y. carried out the sample synthesis, optical measurements and applications with help from B.Z., J.H., and Z.A. All authors contributed to the discussion. The manuscript was written by B.Z. and L.Y. with input from all authors.

## Competing interests

The authors declare no competing interests.
