## [Peer Review File · Nature Communications]

Spatiotemporal control of photochromic upconversion through interfacial energy transferReviewers' comments:

Reviewer #1 (Remarks to the Author):

The manuscript provides interesting results that may be applied for example in anti counterfeiting. The manuscript needs to be read by a native English speaker. This would improve the readability and ease the struggle for the reader of attempting to understand what message the authors are trying to convey.

I have a few concerns which must be addressed by the authors:

The reasoning to include 0.5% Ho³⁺ is never explained.

They briefly mention "energy pooling" to improve the red emission of Er³⁺, but this mechanism is not explained, investigated, nor referenced to previous works in the literature.

Figure 2d reports upconversion spectra with different excitation wavelengths (808 and 1530 nm), but no power densities are reported. The comparison is meaningless without some consideration on excitation source power.

All the temporal-dependent studies (Figure 4) are reported with varying pulse widths, but no power densities at each pulse width is reported. All the trends they observe may simply be a consequence of changing power density when the laser source is modulated.

Also, given how interesting the change in ratio is presented in Figure 4c, the lifetimes of each state should be reported at each of the different pulse widths. The only decay profiles reported are for 0.5 ms and 10 ms. Record the others, and report on the trend in decay times!?

I recommend major revision

Reviewer #2 (Remarks to the Author):

Title: Spatiotemporal control of upconversion photochromic evolution from a single emitter through interfacial energy transfer

Manuscript Number: NCOMMS-23-16708

Authors: Long Yan, Jinshu Huang, Zhengce An, Qinyuan Zhang, and Bo Zhou

Manuscript Summary: The manuscript describes upconversion nanoparticles of a core-shell-shell architecture which leads to enhanced upconversion efficiency and switchable emission colors by the construction of an interfacial energy transfer channel. This was achieved by separating the activator ions and sensitizer ions in core and shell with an inert shell in between.

Reviewers General Comments:

1) The results presented here are somehow novel and of wide interest to the community when one considers the data from Figures 4 and 5. Nevertheless the first part is not new and not easy to follow. Here It remains in large parts unclear what the message is. Therefore, this part needs to be rewritten in a condensed form.

In general, there are some interesting ideas within this publication. The basic concept of using Er@Yb@Y particle architectures is not new, it was for example published recently (2022) by Liu et al. in Nature Communications (<https://doi.org/10.1038/s41467-022-33660-8>). Here, they additionally varied the percentage of Yb and the shell thickness of the Yb shell for this exact particle system, even though it is not surprising from the data reported from other particle systems that an increased active shell layer enhances the luminescence intensity per particle. This is what is basically described in the first part of the manuscript. Very interesting is the concept of inserting an intermediate Y-shell and the investigations about temporal color control. At some points the data description is not concise and the

explanations difficult to follow. The data interpretation needs often to be believed since basic information as solvent or the method which was used for normalization of the luminescence data are missing.

2) The manuscript is too long considering its content and novelty. On the other hand, important data to be able to reproduce this work are missing, especially a detailed characterization of the nanoparticles. Some references on particles of similar structure and are missing, own work is extensively cited, even when not necessarily needed.

3) The logical structure of this manuscript can be improved significantly.

4) Experimental details are not always presented in such a way that this work can be reproduced by any other researchers.

Reviewers Specific Comments:

1) The language needs to be revised, as some sentences are not comprehensive or difficult to understand.

2) The authors need to report the excitation power densities in Figures Fig. S3 and Fig 2d when using 808 nm and 1530 nm excitation.

3) It is curious that only one type of particle architecture – the ones with 20% Yb content - have been fully characterized by showing TEM, XRD and STEM data. Why those have been chosen, others are described as more promising in terms of luminescence efficiency.

4) Please revise Fig. 2e and the description of the results within the text. The axis claims "slope values", which is not clear what is meant exactly by this term. According to the Figure Caption it is believed that the graph displays lifetimes and according to the text (line 96) slopes of power-dependent measurements should be depicted. This is confusing.

5) According to the text in line 103, one cannot understand why the Yb content boosts the green emission. It is mentioned that this was "discussed before". But it is not obvious where this was discussed in detail?

6) The authors explain: "Here we should also note that there exists an optimal thickness of the NaYbF₄ sensitizing layer because it shows a saturation trend when the thickness is over 3.1 nm, might be ascribed to the energy migration over Yb³⁺ sublattice, which would reduce the energy transfer to activators (Fig. S10)". I wonder if this is really a correct interpretation. The saturation seems to be an effect caused by the logarithmic scale, isn't it? The same for the Fig. 3a (line 130).

7) Figure 3d, e: Why are the data points of the same particles at different shell thickness values. This is confusing. I am also wondering why the trends in intensity changes are completely different.

8) Figure 4 depicts an extremely interesting concept. It would be additionally interesting to compare the absolute, but not only the normalized intensities. As it stands for now, the corresponding explanations in the text are quite difficult to understand.

9) Can the authors comment on the role of the Ho³⁺ in the particle. It was never mentioned why Ho was add as additional dopand. Also, controls are missing if Ho will be excluded.

10) Figure 5e: It is recommended not to give the speed in kilometer per second. This seems to be dependent from the diameter ot the rotating unit. Wouldn't it make more sense to give a rotation

frequency in Hz?

Overall, If the authors manage to present a clearer data interpretation and concise explanations of their findings including more information regarding their data, this manuscript should be reconsidered for publishing in Nature Communications.

Reviewer #3 (Remarks to the Author):

The article "Spatiotemporal control of upconversion photochromic evolution from a single emitter through interfacial energy transfer" by Long Yan et al. presents an approach suitable to enhance upconversion luminescence intensity and 'switch' red/green colour by pulse width modulation or increasing pump power intensity of CW photo-excitation.

Although the article is well written, the samples are well prepared and characterized, the figures are clear and interesting, the discussion is reasonable I am not convinced about high / sufficient novelty of the presented studies, which instead I consider as reshaping the already well known effects. Not only the pulse width modulation on the UC colour has been shown before (ref.7 and other not cited here for YbHo, but also YbEr), but spatiotemporal colour 'splitting' was also demonstrated. Moreover, I do not think the discussion provides an 'in-depth insight into the photophysics', but it is rather mechanistic and qualitative description. The UC intensity growth has been also demonstrated by increased sensitizer content as well as by surface passivation with the undoped shell.

Overall, I am of the opinion the manuscript lacks clear and strong novelty.

Reviewer #4 (Remarks to the Author):

In this manuscript, the authors demonstrate bright visible light emission and color tuning from the NaErF₄:Ho(0.5%)@NaYF₄:Yb@NaYF₄ core-shell-shell UCNPs via interfacial energy transfer process. By changing Yb concentration in the NaYF₄:Yb layer and the thickness of the NaYF₄:Yb layer, highly luminescent UCNPs could be obtained. The modulation of the pulse width and the power of the incident laser enabled the fine tuning of the emission color from the core-shell-shell UCNPs and green to red color change was shown. By utilizing these unique properties, anti-counterfeiting patterns were successfully demonstrated. Also, the authors converted temporal photochromic evolution to spatial color distribution and they utilized this for speed monitoring. These results are quite interesting and the reviewer thinks that this study is interesting to the broad readership in Nature Communications. However, there are many flaws which should be addressed. In some parts, figure captions are not consistent with figures and the figure is not consistent with explanation in the main text as follows. These should be carefully checked, and following issues should be addressed before acceptance.

1. While Fig. 2e shows slope values dependent on Yb³⁺ concentrations in NaYF₄:Yb layer, Fig. 2e caption described the dependence of lifetime values of Er³⁺ emissions on Yb³⁺ concentration. Fig. 2e caption seems to be wrong.

2. In page 6, it is described that the presence of Yb³⁺ can further improve the green emission and consequently promote the CR. How did the presence of Yb³⁺ in the shell improve green emission from the Er³⁺ in the core and promote the CR? According to the previous study, CR in Er³⁺ increases when the Er³⁺ concentration increases in the core (Angew. Chem. Int. Ed. 2017, 2017, 56, 7605 and Nanoscale, 2017, 9, 7941). If the concentration of Yb³⁺ increases in the NaErF₄:Ho(5%) core, the presence of Yb³⁺ would result in the decrease of Er³⁺ amounts and increase the distance between Er³⁺ ions in the core. Then, CR $[(2H_{11/2}, 4S_{3/2}); 4I_{9/2}] \rightarrow [4F_{9/2}; 4F_{9/2}]$ will decrease and green

emission will be improved. However, in this study, Yb³⁺ is present in the shell and thus, related explanation need to be provided. In addition, if the presence of Yb³⁺ can further improve the green emission, it will suppress the CR. As the authors explained in page 6, stronger red emission than green emission is due to CR among Er³⁺ ions.

Thus, the sentence "the presence of Yb³⁺ can further improve the green emission and consequently promote the CR" does not seem to make sense.

3. The authors explained that the presence of Yb³⁺ can further improve the green emission and consequently promote the CR, and this provides a new channel to help the population of the red emitting level and reduce the slope value. If there is a new channel affects the PL intensity, decay and/or rise behavior might change. Thus, the results can be explained with time-resolved PL spectra. In Fig. 2c, PL intensity of the NaErF₄:Ho(5%)@NaYF₄:Yb(x)@NaYF₄ core-shell-shell UCNPs increased with increasing Yb³⁺ concentration. However, decay time of the core-shell-shell UCNPs showed increased and then decreased as the Yb concentration increased in Fig. S11 and Table S1. Why did the decay time show different trend from the PL intensity change with varying Yb concentration?

4. When the Fig. 3d and 3e, NaYF₄ interlayer thickness values are not consistent. Moreover, the PL spectra data (Fig. 3d) are not consistent with normalized intensity (Fig. 3e) of NaErF₄:Ho(5%)@NaYF₄@NaYbF₄@NaYF₄ UCNPs with varying NaYF₄ interlayer thickness. The figure and data should be clear.

5. In this study, NaErF₄:Ho(5%)@NaYF₄@NaYbF₄@NaYF₄ with NaYF₄ interlayer thickness of 2.1 nm showed the highest PL intensity. However, the authors did not investigate the color tuning for the NaErF₄:Ho(5%)@NaYF₄(2.1 nm)@NaYbF₄@NaYF₄ UCNPs. Also, the authors did not use the NaErF₄:Ho(5%)@NaYF₄(2.1 nm)@NaYbF₄@NaYF₄ UCNPs for optical applications. Why?

6. The NaErF₄:Ho(5%)@NaYF₄@NaYbF₄@NaYF₄ UCNPs showed PL enhancement and then decrease of the PL intensity with increasing the NaYF₄ interlayer thickness. According to the authors, it is due to effective depression of energy loss due to backward energy transfer channels. The presence of NaYF₄ interlayer between NaErF₄:Ho and NaYbF₄ may depress energy transfer from Yb³⁺ to Er³⁺ and backward energy transfer from Er³⁺ in the core to Yb³⁺ in the NaYbF₄ shell. More explanation as to why the backward energy transfer was more effectively suppressed needs to be provided.

7. In Fig. 3, green and red emission intensities increased when NaYF₄ interlayer thickness reached 2.1 nm and then it decreased when the NaYF₄ interlayer thickness was larger than 2.1 nm. The decay time measured for green emission showed the same trend. However, decay time measured for red emission increased when the NaYF₄ interlayer thickness reached 4.3 nm and it was shortened when the NaYF₄ interlayer thickness was 8.7 nm. Unlike the green emission, for red emission, why is the trend in the PL intensity change different from that in the decay time change?

8. Fig. S12(b-f) shows TEM images of NaErF₄:Ho(5%)@NaYF₄ core-shell UCNPs with increased thickness of NaYF₄ layer. According to the labeling, it is expected that the sizes of core-shell UCNPs correspond to 15.3, 16.5, 18.5, 22.9, and 31.1 nm for (b), (c), (d), (e), and (f), respectively. However, the sizes of the core-shell UCNPs in (a), (c), and (e) look very similar. In addition, brightness contrast between core and shell regions is expected to be shown due to large difference of atomic numbers of Er in the core and Y in the shell. However, the brightness contrast is only observed in Fig. S12f. Is it due to thin shell thickness for Fig. S12(b-e)?

9. In page 5, 4S_{2/3} → 4I_{15/2} should be 4S_{3/2} → 4I_{15/2}. In page 11, red-emitting 4F_{7/2} should be 4F_{9/2}.

10. In page 11, it is described that the population of Er³⁺ at the red emitting 4F_{7/2} level can be improved with a decline of the intensity-power slope values (Fig. 2f). However, Fig. 2f shows PL spectra of NaErF₄:Ho(0.5%)@NaYbF₄@NaYF₄ UCNPs with varying NaYbF₄ thickness. How did the Fig.

2f show a decline of the intensity-power slope values?

11. The authors claimed that emission intensity was enhanced by IET from NaYbF₄ to NaErF₄:Ho(5%). Then, what are upconversion quantum yields (UCQYs) of NaErF₄:Ho(5%)@NaYbF₄@NaYF₄ UCNPs. Is there a power density dependence of UCQYs?

12. In Fig. 5 caption, NaErF₄(0.5mol%)-NaYF₄:Yb(20 mol%)-NaYF₄ UCNPs seems to be NaErF₄:Ho(0.5 mol%)-NaYF₄:Yb(20 mol%)-NaYF₄. It should be checked. Why did the authors used NaErF₄:Ho(0.5 mol%)-NaYF₄:Yb(20 mol%)-NaYF₄ UCNPs for applications although NaErF₄:Ho(0.5 mol%)-NaYF₄:Yb(10 mol%)-NaYF₄ UCNPs showed higher G/R ratio than NaErF₄:Ho(0.5 mol%)-NaYF₄:Yb(20 mol%)-NaYF₄ UCNPs?

13. The authors claim that the maximum speed sensitivity was much higher than the recent reports. However, in ref. 49, the highest Sa value is 0.045 s/cm. On the other hand, in this study, Sa is 1.29 (km/h)⁻¹, which is the same as 0.0469 s/cm. Thus, Sa (1.29 h/km) in this study is not much higher than the previous study.

14. In this study, very interesting results were shown in Fig. 5. However, related explanation is not enough. For example, an upper left image (red square) in Fig. 5b showed color change from red for long pulse excitation to mixed color of red and green in QR code for short pulse excitation. That is, in some regions, red color changed to green color and in other regions, red color was maintained. Then, were different compositions of the UCNPs patterned together for the image shown in Fig. 5b? Since the authors showed that core-shell-shell UCNPs showed green and red emission for short and long pulse excitations, respectively, mixed color of red and green shown in the upper right image of Fig. 5b needs to be explained in more detail.

Point-by-Point Responses to the comments of Reviewers 1~4

Manuscript Number: NCOMMS-23-16708

Title: Spatiotemporal control of upconversion photochromic evolution from a single emitter through interfacial energy transfer

(Changes in the revised manuscript as a response to the reviewers' comments are highlighted in red color.)

Responses to Reviewer #1

Reviewer #1: The manuscript provides interesting results that may be applied for example in anti counterfeiting. This would improve the readability and ease the struggle for the reader of attempting to understand what message the authors are trying to convey.

Reply: We are very grateful to this reviewer for the positive comments and suggestions.

Q1. *The reasoning to include 0.5% Ho³⁺ is never explained. They briefly mention "energy looping" to improve the red emission of Er³⁺, but this mechanism is not explained, investigated, nor referenced to previous works in the literature.*

Reply: In the present manuscript, Ho³⁺ was also selected to manipulate the upconversion process of Er³⁺. Namely, the populations of Er³⁺ between its ⁴I_{11/2} and ⁴I_{13/2} can be tuned through energy transfer looping of ⁴I_{11/2} (Er³⁺) → ⁵I₆ (Ho³⁺) → ⁴I_{13/2} (Er³⁺) and the red upconversion channel (⁴I_{13/2} → ⁴F_{9/2}) is markedly promoted, resulting in an enhancement of red emission under 980 nm excitation. Here the ref 44 (Nanoscale, 2018, 10, 17949-17957) was cited for reference. Especially for the design of NaErF₄:Ho(0.5%)@NaYbF₄@NaYF₄ core-shell-shell nanostructure, the NaYbF₄ interlayer could absorb the excitation energy more efficiently and transfer it to Er³⁺, leading to further enhanced red emission (see the newly added Supplementary Fig. 1 in the revised SI). In addition, the sample with doping of Ho³⁺ showed much longer rise time and decay time of the red emission, providing an opportunity for dynamic control of photochromic evolution.

In the revised version (pp4), the following sentence was added for a clear explanation: "The small amount of Ho³⁺ in NaErF₄ lattice is used to tune the populations of Er³⁺ at its ⁴I_{11/2} and ⁴I_{13/2} states through energy looping from Er³⁺ (⁴I_{11/2}) to Ho³⁺ (⁵I₆) and then back to Er³⁺ (⁴I_{13/2})⁴⁴."

The newly added Supplementary Fig. 1:

Supplementary Fig. 1. (a) Schematic of upconversion processes and energy transfer looping (ET1+ET2) from Er³⁺ (⁴I_{11/2}) to Ho³⁺ (⁵I₆) and then back to Er³⁺ (⁴I_{13/2}) in the proposed design of NaErF₄:Ho(0.5 mol%)@NaYbF₄@NaYF₄ core-shell-shell nanostructure upon 980 nm excitation. (b) Upconversion emission spectra of NaErF₄:Ho(0.5 mol%)@NaYbF₄@NaYF₄ (ErHo@Yb@Y) and NaErF₄@NaYbF₄@NaYF₄ (Er@Yb@Y) core-shell-shell nanoparticles under 980 nm excitation. (c) Time-dependent emission profiles of Er³⁺ at 654 nm from (b) samples.

Q2. Figure 2d reports upconversion spectra with different excitation wavelengths (808 and 1530 nm), but no power densities are reported. The comparison is meaningless without some consideration on excitation source power.

Reply: Thanks for the advice. In Fig. 2d, all excitation power densities for 980, 808, 1530 nm excitations were fixed to 11.6 W cm⁻², and the intensity of sample without Yb³⁺ content in intermediate layer was normalized to 1 in order to compare the intensity trend with increasing Yb³⁺ under different excitations. We have supplemented the excitation power density to Fig. 2 caption by adding the " All excitation power densities were 11.6 W cm⁻²."

Q3. All the temporal-dependent studies (Figure 4) are reported with varying pulse widths, but no power densities at each pulse width is reported. All the trends they observe may simply be a consequence of changing power density when the laser source is modulated.

Author Reply: When the laser source was modulated, the instantaneous excitation power densities with different pulse widths were fixed to a constant to avoid the interference of power factor. The steady and instantaneous excitation power densities were all fixed to 11.6 W cm⁻². And we have supplemented the excitation power density to Fig. 4 caption by adding the " All excitation power densities were 11.6 W cm⁻²."

Q4. Also, given how interesting the change in ratio is presented in Figure 4c, the lifetimes of each state should be reported at each of the different pulse widths. The only decay profiles reported are for 0.5 ms and 10 ms. Record the others, and report

on the trend in decay times!?

Reply: Thanks for the valuable advice. As previously reported (Ref 27: Angew. Chem. Int. Ed. 2022, 61, e202212089), the decay time of Er-based nanoparticles could be modulated by excitation pulse width due to the energy migration when the particle size is larger than 20 nm. In our manuscript, the decay time of NaErF₄:Ho(0.5%)@NaYF₄:Yb(10%)@NaYF₄ nanoparticles at 540 and 654 nm were measured under different pulse widths as shown in the newly added Supplementary Fig. 31 in SI). The decay times of 540 and 654 nm show an increase with the increase of pulse width before a saturation trend, which is attributed to the energy migration among Yb³⁺ ions because of its much higher doping content. And ref 27 was cited here. More importantly, the lifetime of red emission is always longer than that of green emission under different pulse width excitations.

In the revised manuscript (pp 12), we added the following sentences for a deeper discussion: “A tuning of excitation pulse widths imposes an impact on the decay curves, and longer lifetime at wider pulse width may be owing to the possible energy migration among Yb³⁺ ions (Supplementary Fig. 31) ²⁷. While it should be noted that the lifetime of red emission is always longer than that of green emission under different pulse width excitations.”.

The newly added Supplementary Fig. 31:

Supplementary Fig. 31. (a,b) Decay curves of Er³⁺ at its (a) ⁴S_{3/2} (540 nm) and (b) ⁴F_{9/2} (654 nm) from NaErF₄:Ho(0.5 mol%)@NaYF₄:Yb(10 mol%)@NaYF₄ core-shell-shell nanoparticles under 980 nm excitation with different pulse widths (0.5-10 ms). (c) Lifetime values from (a) and (b).

Responses to Reviewer #2

Reviewer #2: The manuscript describes upconversion nanoparticles of a core-shell-shell architecture which leads to enhanced upconversion efficiency and switchable emission colors by the construction of an interfacial energy transfer channel. This was achieved by separating the activator ions and sensitizer ions in core and shell with an inert shell in between.

Overall, If the authors manage to present a clearer data interpretation and concise explanations of their findings including more information regarding their data, this manuscript should be reconsidered for publishing in Nature Communications.

Reply: We are very grateful to the reviewer for the positive comments and

suggestions. And we have re-organized the logical structure properly and rewritten the manuscript according to the reviewer's suggestions.

General Comments:

Q1. *The results presented here are somehow novel and of wide interest to the community when one considers the data from Figures 4 and 5. Nevertheless the first part is not new and not easy to follow. Here It remains in large parts unclear what the message is. Therefore, this part needs to be rewritten in a condensed form.*

In general, there are some interesting ideas within this publication. The basic concept of using Er@Yb@Y particle architectures is not new, it was for example published recently (2022) by Liu et al. in Nature Communications (<https://doi.org/10.1038/s41467-022-33660-8>). Here, they additionally varied the percentage of Yb and the shell thickness of the Yb shell for this exact particle system, even though it is not surprising from the data reported from other particle systems that an increased active shell layer enhances the luminescence intensity per particle. This is what is basically described in the first part of the manuscript. Very interesting is the concept of inserting an intermediate Y-shell and the investigations about temporal color control. At some points the data description is not concise and the explanations difficult to follow. The data interpretation needs often to be believed since basic information as solvent or the method which was used for normalization of the luminescence data are missing.

Reply: The research on core-shell based UCNPs has been an attractive hotspot over the past decade. In particular, the core-multi-shell structure nanoparticles were widely designed enabling adjustable luminescence properties based on EMU (Energy migration upconversion) or IET (Interfacial energy transfer) model. Recently, some reports showed that coating a NaYbF₄ shell in UCNPs would promote their upconversion intensity via IET, such as the design of NaLuF₄:Er@NaYbF₄@NaLuF₄ in the paper suggested by the reviewer (Nature Communications, 2022, 13, 5927. <https://doi.org/10.1038/s41467-022-33660-8>; cited as ref 48). However, those works were mainly focused on the luminescence enhancement. Here in our work we designed a core-multi-shell structure consisting of Er and Yb lattices instead of low concentration doping to tune the energy transfer between Er and Yb sublattices and control their upconversion property. The presence of Yb shell layer indeed leads to enhanced upconversion. And moreover, through controlling IET from Yb³⁺ to Er³⁺, the upconversion dynamics can be further manipulated with resultant photochromic evolution (e.g., color change from red to green or vice versa) by spatial or temporal control of the ionic interactions. Besides, we also observed the back energy transfer from Er³⁺ to Yb³⁺ which is also important to tune emission colors. The ionic spatial modulation including concentration of Yb³⁺, thickness of NaYbF₄ layer and ionic separation by a NaYF₄ interlayer were key factors to realize photochromic evolution in time domain. Thus, the first part is necessary to present a detailed investigation and highlight the importance of our design.

In the revised version, we modified the content properly and supplemented more experimental details including synthesis method for each shell layer, accurate material ratios, dispersion solvent, TEM images, XRD patterns (see the newly added/updated

Supplementary Figs. 2-3, 8-9, 15-16; shown in the response to the next comment **Q2**) in SI to make it easier to understand. Also, the method used for normalizing the luminescence data was also supplemented in the figure captions (including Supplementary Figs. 1, 21, 24, 26, 27, and 29).

Q2. *The manuscript is too long considering its content and novelty. On the other hand, important data to be able to reproduce this work are missing, especially a detailed characterization of the nanoparticles. Some references on particles of similar structure and are missing, own work is extensively cited, even when not necessarily needed.*

Reply: Thanks for your valuable suggestion. In this manuscript, each section is important to fully demonstrate our experimental results and discussions. The spatial distribution of lanthanides not only plays a key role in steady-state upconversion enhancement, but also in the temporal control of emission colors. The first section demonstrated how to optimize the upconversion in the nanostructure by designing the sensitizing interlayer with tuning Yb content and its shell thickness via IET. Then the steady-state upconversion process and the control of IET process under spatial scale are presented. Next, the upconversion channels were discussed for dynamic control of photochromic evolution. Last, the multi-mode properties with photochromic evolution were applied in different photonics applications. For a concise and easy understanding, we have properly revised the manuscript by rewriting or deleting some sentences. We believe the revised version presents a concise and logical demonstration of our idea and experimental results.

Regarding the detailed characterization of nanoparticles, they are supplemented in SI to ensure repeatability of this work. These include the TEM and XRD results of NaErF₄:Ho@NaYF₄:Yb(0-100 mol%)@NaYF₄ UCNPs, NaErF₄:Ho@NaYbF₄@NaYF₄ UCNPs with different NaYbF₄ layer thicknesses, and NaErF₄:Ho@NaYF₄@NaYbF₄@NaYF₄ UCNPs with different NaYF₄ interlayer thicknesses (see the newly added/updated Supplementary Figs. 2-3, 8-9, 15-16). The spectra with recorded initial intensity under dynamic excitation were also provided in SI (see the newly added Supplementary Figs. 22, 24, 26-27, 29). The related references were also supplemented (see the refs 2, 27-29, 31, 35, 49, 51) and several refs in the original version were deleted.

The newly added Supplementary Figs. 2-3, 8-9, 15-16, 22, 24, 26-27, 29:

Supplementary Fig. 2. (a-c) TEM images of $\text{NaErF}_4:\text{Ho}(0.5 \text{ mol}\%)\text{@NaYF}_4:\text{Yb}(0\text{-}100 \text{ mol}\%)\text{@NaYF}_4$ core-shell-shell nanoparticles at each stage. Core is $\text{NaErF}_4:\text{Ho}(0.5 \text{ mol}\%)$. CS and CSS represent $\text{NaErF}_4:\text{Ho}(0.5 \text{ mol}\%)\text{@NaYF}_4:\text{Yb}(0\text{-}100 \text{ mol}\%)$ core-shell and $\text{NaErF}_4:\text{Ho}(0.5 \text{ mol}\%)\text{@NaYF}_4:\text{Yb}(0\text{-}100 \text{ mol}\%)\text{@NaYF}_4$ core-shell-shell nanoparticles, respectively.

Supplementary Fig. 3. XRD patterns of $\text{NaErF}_4:\text{Ho}(0.5 \text{ mol}\%)\text{@NaYF}_4:\text{Yb}(0\text{-}100 \text{ mol}\%)\text{@NaYF}_4$ core-shell-shell nanoparticles at each stage. Core is $\text{NaErF}_4:\text{Ho}(0.5 \text{ mol}\%)$. CS and CSS represent $\text{NaErF}_4:\text{Ho}(0.5 \text{ mol}\%)\text{@NaYF}_4:\text{Yb}(0\text{-}100 \text{ mol}\%)\text{@NaYF}_4$ core-shell and $\text{NaErF}_4:\text{Ho}(0.5 \text{ mol}\%)\text{@NaYF}_4:\text{Yb}(0\text{-}100 \text{ mol}\%)\text{@NaYF}_4$ core-shell-shell nanoparticles, respectively.

Supplementary Fig. 8. (a-c) TEM images of $\text{NaErF}_4:\text{Ho}(0.5 \text{ mol}\%)\text{@NaYbF}_4\text{@NaYF}_4$ core-shell-shell nanoparticles with increasing NaYbF_4 layer

thickness (0.4-7.5 nm). Core is NaErF₄:Ho(0.5 mol%). CS and CSS represent NaErF₄:Ho(0.5 mol%)@NaYbF₄ core-shell and NaErF₄:Ho(0.5 mol%)@NaYbF₄@NaYF₄ core-shell-shell nanoparticles. (d) Corresponding particle size of the NaErF₄:Ho(0.5 mol%) core and coated with different NaYbF₄ interlayer thicknesses.

Supplementary Fig. 9. XRD patterns of NaErF₄:Ho(0.5 mol%)@NaYbF₄@NaYF₄ core-shell-shell nanoparticles with increasing NaYbF₄ layer thickness (0-3.1 nm). Core is NaErF₄:Ho(0.5 mol%). CS and CSS represents NaErF₄:Ho(0.5 mol%)@NaYbF₄ core-shell and NaErF₄:Ho(0.5 mol%)@NaYbF₄@NaYF₄ core-shell-shell nanoparticles.

Supplementary Fig. 15. (a-d) TEM images of NaErF₄:Ho(0.5

mol%)@NaYF₄@NaYbF₄@NaYF₄ core-multishell nanoparticles with increasing NaYbF₄ layer thickness at each stage. C, CS, CSS and CSSS represents the NaErF₄:Ho(0.5 mol%) core, NaErF₄:Ho(0.5 mol%)@NaYF₄ core-shell, NaErF₄:Ho(0.5 mol%)@NaYF₄@NaYbF₄ core-shell-shell and NaErF₄:Ho(0.5 mol%)@NaYF₄@NaYbF₄@NaYF₄ core-multishell nanoparticles with different NaYF₄ inter layer thicknesses (0.5-8.7 nm), respectively. (e) Corresponding particle size of the NaErF₄:Ho(0.5 mol%) core and coated with different NaYF₄ interlayer thicknesses.

Supplementary Fig. 16. XRD patterns of NaErF₄:Ho(0.5 mol%)@NaYF₄@NaYbF₄@NaYF₄ nanoparticles with increasing NaYbF₄ layer thickness (0-8.7 nm). Core is NaErF₄:Ho(0.5 mol%). CS, CSS and CSSS represent NaErF₄:Ho(0.5 mol%)@NaYF₄ core-shell, NaErF₄:Ho(0.5 mol%)@NaYF₄@NaYbF₄ core-shell-shell and NaErF₄:Ho(0.5 mol%)@NaYF₄@NaYbF₄@NaYF₄ core-multishell nanoparticles.

Supplementary Fig. 22. (a-h) Non-steady state upconversion emission spectra of (a) NaErF₄:Ho(0.5 mol%)@NaYF₄ core-shell and (b-h) NaErF₄:Ho(0.5 mol%)@NaYF₄:Yb(1,2,5,10,20,60,100 mol%)@NaYF₄ core-shell-shell nanoparticles under pulse 980 nm excitation with frequency of 50 Hz.

Supplementary Fig. 24. Non-steady state upconversion emission spectra of the NaErF₄:Ho(0.5 mol%)@NaYbF₄@NaYF₄ core-shell-shell nanoparticles under pulse 980 nm excitation with frequency of 50 Hz. The thickness of NaYbF₄ layer was precisely controlled from 0 to 7.5 nm.

Supplementary Fig. 26. Non-steady state upconversion emission spectra of the NaErF₄:Ho(0.5 mol%)/NaYF₄/NaYbF₄/NaYF₄ core-shell-shell nanoparticles under pulse 980 nm excitation with frequency of 50 Hz. The thickness of NaYF₄ interlayer was precisely controlled from 0 to 8.7 nm.

Supplementary Fig. 27. (a,b) Normalized non-steady state upconversion emission spectra of (a) NaErF₄@NaYF₄:Yb(20 mol%)/NaYF₄ and (b) NaErF₄@NaYbF₄/NaYF₄ core-shell-shell nanoparticles under pulse 980 nm excitation with frequency of 50 Hz. (c,d) Corresponding recorded emission spectra of (a) and (b).

Supplementary Fig. 29. (a,b) Normalized non-steady state upconversion emission spectra of (a) NaErF₄@NaYF₄ and (b) NaErF₄:Ho(0.5 mol%)@NaYF₄ core-shell nanoparticles under pulse 980 nm excitation with frequency of 50 Hz. (c,d) Corresponding recorded emission spectra of (a) and (b).

The newly cited refs 2, 27-29, 31, 35, 49, 51:

(2) Ou, Y., et al. Host differential sensitization toward color/lifetime-tuned lanthanide coordination polymers for optical multiplexing. *Angew. Chem. Int. Ed.* **2020**, 59(52), 23810-23816.

(27) Han, Y., et al. Modulating the rise and decay dynamics of upconversion luminescence through controlling excitations. *Angew. Chem. Int. Ed.* **2022**, 61(45), e202212089.

(28) Dawson, P. & Romanowski, M. Excitation modulation of upconversion nanoparticles for switch-like control of ultraviolet luminescence. *J. Am. Chem. Soc.* **2018**, 140(17), 5714-5718.

(29) Labrador-Paez, L., et al. Excitation pulse duration response of upconversion nanoparticles and its applications. *J. Phys. Chem. Lett.* **2022**, 13(48), 11208-11215.

(31) McLellan, C. A., et al. Engineering bright and mechanosensitive alkaline-earth rare-earth upconverting nanoparticles. *J. Phys. Chem. Lett.* **2022**, 13(6), 1547-1553.

(35) Liu, X., et al. Fast wide-field upconversion luminescence lifetime thermometry enabled by single-shot compressed ultrahigh-speed imaging. *Nat. Commun.* **2021**, 12, 6401.

(49) Blasse, G. Energy transfer in oxidic phosphors. *Phys. Lett.* **1968**, 28A(6), 444-445. Wen, S., et al. Advances in highly doped upconversion nanoparticles. *Nat. Commun.* **2018**, 9(1), 2415.

(51) Sun, T., et al. Ultralarge anti-stokes lasing through tandem upconversion. *Nat. Commun.* **2022**, 13(1), 1032.

Q3. The logical structure of this manuscript can be improved significantly.

Reply: The revised manuscript has re-organized the logical structure properly according to the suggestions. The revised manuscript is in the following logical

structure: Firstly, we focused on enhancement and colour tuning for the design of NaErF₄:Ho@NaYbF₄@NaYF₄ core-shell-shell nanostructure. Secondly, we described a new strategy based on the spatial control between Yb³⁺ and Er³⁺ sublattices to further enhance and manipulate the ionic interactions between Er³⁺ and Yb³⁺. Thirdly, the temporal control on upconversion photochromic evolution was discussed systematically. Finally, we demonstrated the frontier applications based on the versatile luminescent properties of our designed nanoparticles.

Q4. *Experimental details are not always presented in such a way that this work can be reproduced by any other researchers.*

Reply: Thanks for the advice. In the revised version, more detailed experimental details have been supplemented, including the synthesis methods of NaErF₄:Ho@NaYF₄:Yb(0-100 mol%)@NaYF₄ nanoparticles with different Yb³⁺ doping concentrations, NaErF₄:Ho@NaYbF₄@NaYF₄ nanoparticles with different NaYbF₄ thicknesses, and NaErF₄:Ho@NaYF₄@NaYbF₄@NaYF₄ nanoparticles with different NaYF₄ interlayer thicknesses, NaYbF₄:Er@NaYF₄, and NaYF₄:Yb/Er@NaYF₄ nanoparticles, see the revised experimental procedures (2. Synthesis of nanoparticles and 4. Application of colour-switchable upconversion) and newly added Supplementary Figs 2-3, 8-9, 15-16, 22, 24, 26-27 and 29. We hope these new data help the reproduction by any other researchers effectively.

Specific Comments:

Q1. *The language needs to be revised, as some sentences are not comprehensive or difficult to understand.*

Reply: Thanks for the advice. The language has been revised and polished thoroughly in the revised version.

Q2. *The authors need to report the excitation power densities in Figures Fig. S3 and Fig 2d when using 808 nm and 1530 nm excitation.*

Reply: In Fig. 2d and Fig. S3, the power densities for 808 and 1530 nm excitations were fixed to 11.6 W cm⁻², and this was added in the figure captions.

Q3. *It is curious that only one type of particle architecture – the ones with 20% Yb content - have been fully characterized by showing TEM, XRD and STEM data. Why those have been chosen, others are described as more promising in terms of luminescence efficiency.*

Reply: Thanks for the valuable advice. In the original version the NaErF₄:Ho(0.5%)@NaYF₄:Yb(20%)@NaYF₄ nanoparticles were selected as a representative for detailed characterization due to their properties such as enhanced luminescence intensity, dynamic control of photochromic evolution, power-induced emission discoloration. Here in the revised version, we further supplemented the TEM and XRD data of NaErF₄:Ho@NaYF₄:Yb(0-100 mol%)@NaYF₄ nanoparticles with different Yb³⁺ doping concentrations, NaErF₄:Ho@NaYbF₄@NaYF₄ nanoparticles with different NaYbF₄ thicknesses, and NaErF₄:Ho@NaYF₄@NaYbF₄@NaYF₄

nanoparticles with different NaYF₄ interlayer thicknesses in SI, see the newly added/updated Supplementary Figs. 2-3, 8-9, 15-16, 22, 24, 26-27, and 29. These data should present a more full and detailed characterization of the samples of our work.

Q4. Please revise Fig. 2e and the description of the results within the text. The axis claims “slope values”, which is not clear what is meant exactly by this term. According to the Figure Caption it is believed that the graph displays lifetimes and according to the text (line 96) slopes of power-dependent measurements should be depicted. This is confusing.

Reply: Thanks for the careful reading. This confuse is caused by the incorrect description in the Fig. 2e caption that "lifetime" should be "slope values". In Fig. 2e, the y axis of “slope values” is correct. Sorry for this mistake and we have corrected it in revised manuscript.

Q5. According to the text in line 103, one cannot understand why the Yb content boosts the green emission. It is mentioned that this was "discussed before". But it is not obvious where this was discussed in detail?

Reply: Thanks for the question. The "discussed before" means the sentence in lines 87-88 in the original version (“Note that the red emission is much greater than that of the green emission, as a result of cross relaxations among Er³⁺ sublattice such as [(²H_{11/2}, ⁴S_{3/2}); ⁴I_{9/2}] \rightarrow [⁴F_{9/2}; ⁴F_{9/2}].”) According to the text in line 84 “With the increase of Yb³⁺ concentration in the sensitizing interlayer, the upconversion emission intensity shows a significant monotonous enhancement”, we discussed the enhancement of both green and red emission with increasing Yb³⁺ content. As shown in Fig. 3a, the green and red emissions in NaErF₄:Ho(0.5%)@NaYbF₄@NaYF₄ nanoparticles are enhanced by 151.4 and 10.7 folds, respectively, compared to the NaErF₄:Ho(0.5%)@NaYF₄@NaYF₄ control sample. This is resulted from the interfacial energy transfer from Yb³⁺ to Er³⁺. While due to the occurrence of cross relaxations such as [(²H_{11/2}, ⁴S_{3/2}); ⁴I_{9/2}] \rightarrow [⁴F_{9/2}; ⁴F_{9/2}], many Er³⁺ ions in the ²H_{11/2}, ⁴S_{3/2} levels were depopulated to the lower-lying red-emitting ⁴F_{9/2} level. As a result, the red emission is much greater than that of the green emission.

For a clearer discussion, in the revised version (pp 6), the relevant sentence was revised to be "The presence of Yb³⁺ in the interlayer can further improve the population of Er³⁺ at its green emitting levels through IET and consequently promote the cross relaxation of [(²H_{11/2}, ⁴S_{3/2}); ⁴I_{9/2}] \rightarrow [⁴F_{9/2}; ⁴F_{9/2}]. This adds a new channel to increase the population of red emitting ⁴F_{9/2} level and reduce the slope value."

Q6. The authors explain: “Here we should also note that there exists an optimal thickness of the NaYbF₄ sensitizing layer because it shows a saturation trend when the thickness is over 3.1 nm, might be ascribed to the energy migration over Yb³⁺ sublattice, which would reduce the energy transfer to activators (Fig. S10)”. I wonder if this is really a correct interpretation. The saturation seems to be an effect caused by the logarithmic scale, isn't it? The same for the Fig. 3a (line 130).

Reply: Thanks for your question. In Fig S10 in original SI, we showed the logarithmic

scale of the normalized upconversion intensity for an easy comparison of the green and red emission enhancement because the green emission is much weaker than the red emission. As shown in the following Supplementary Fig. 12a,b, we converted the y-axis to linear presentation for both absolute and normalized intensity. We can see that the absolute intensity of both green and red emissions indeed show a saturation trend when the NaYbF₄ layer reaches 3.1 nm. This might be due to the possible energy migration over Yb³⁺ sublattice. In this case the energy in the ²F_{5/2} of Yb³⁺ can be lost by energy migration instead of transferring to Er³⁺. The saturation trend for Fig 3a with a tuning of Yb³⁺ content is also easily observed for the linear scale of the intensity, see Supplementary Fig. 12c,d. Here for a clearer comparison we also added the data of the total upconversion intensity of green and red emission in the updated Supplementary Fig. 12 in the revised SI. Therefore, the saturation trend is not by the logarithmic scale plot of the intensity data. We replaced the Supplementary Fig. 12 in the original version by this figure in the revised SI to highlight this trend.

The updated Supplementary Fig. 12:

Supplementary Fig. 12. (a) Upconversion enhancement factor and (b) recorded upconversion emission intensity from the NaErF₄:Ho(0.5 mol%)@NaYbF₄@NaYF₄ core-shell-shell nanoparticles with thicknesses of NaYbF₄ interlayer from 0 to 7.5 nm. (c) Upconversion enhancement factor and (d) recorded upconversion emission intensity from the NaErF₄:Ho(0.5 mol%)@NaYF₄:Yb(0-100 mol%)@NaYF₄ core-shell-shell nanoparticles with Yb³⁺ concentrations in interlayer from 0 to 100 mol%.

Q7. Why are the data points of the same particles at different shell thickness values. This is confusing. I am also wondering why the trends in intensity changes are completely different.

Reply: Thanks for the question. Such confuse was caused by the mistake in the plot of the emission profiles in Fig 3d (it was the data in Fig 2f in the original version by mistake). We revised this by using the correct data, and the normalized intensity dependences on NaYF₄ layer thickness is correct. Such correction should remove the misunderstanding.

The updated Fig. 3c (without any change in its caption):

Q8. Figure 4 depicts an extremely interesting concept. It would be additionally interesting to compare the absolute, but not only the normalized intensities. As it stands for now, the corresponding explanations in the text are quite difficult to understand.

Reply: Thank you for your high evaluation to our manuscripts. We have supplemented the corresponding data of recorded absolute intensity in SI, see Supplementary Figs. 22, 24, 26, 27c,d and 29c,d. The reduction of the upconversion in recorded emission spectra is due to the reduced irradiation time as the pulse width decreases. And the descriptions and explanations were also properly revised for an easier understanding.

Q9. Can the authors comment on the role of the Ho³⁺ in the particle. It was never mentioned why Ho was add as additional dopant. Also, controls are missing if Ho will be excluded.

Reply: Thanks for the advice. In the revised version, we have added the role of Ho³⁺ in the NaErF₄ lattice, which can modulate the populations of Er³⁺ between its ⁴I_{11/2} and ⁴I_{13/2} states through energy transfer looping of ⁴I_{11/2} (Er³⁺) → ⁵I₆ (Ho³⁺) → ⁴I_{13/2} (Er³⁺) and improve the red emission of Er³⁺ (see Nanoscale, 2018, 10, 17949-17957; ref 44). In the revised version (pp 4), the following sentence was added for a clear explanation: "The small amount of Ho³⁺ in NaErF₄ lattice is used to facilitate the populations of Er³⁺ from its ⁴I_{11/2} to ⁴I_{13/2} states through energy looping from Er³⁺ (⁴I_{11/2}) to Ho³⁺ (⁵I₆) and then back to Er³⁺ (⁴I_{13/2})⁴⁴." And a new Supplementary Fig. 1 was added in SI. This response can also refer to the answer to the comment **Q1** of reviewer #1.

Q10. Figure 5e: It is recommended not to give the speed in kilometer per second. This

seems to be dependent from the diameter of the rotating unit. Wouldn't it make more sense to give a rotation frequency in Hz?

Reply: In this section, the laser spot irradiated on a fixed point and sample passes through the spot with the rotation of the bottom circular tray. At the same rotation speed (fixed frequency), the linear velocity at different positions in the radius direction is different, resulting in different time of sample passing through laser spot region (pulse width). In this case, the pulse width occupied the main factors on emission colour change. With known spot size, we chose the linear velocity as a parameter of dynamic control. The corresponding rotation rate, frequency, linear speed and angular speed are supplemented as following newly added as Supplementary Table 6 in SI).

Supplementary Table 6. A summary of rotation rate, frequency, linear speed and angular speed for the velocity monitoring experiment.

Rotation rate (r/min)	Rotation frequency (Hz)	Linear speed (km/h)	Angular speed (rad/s)
120	2.0	2.3	4.0 π
200	3.3	3.7	6.6 π
300	5.0	5.7	10 π
400	6.7	7.5	13.4 π
600	10.0	11.3	20.0 π
800	13.3	15.1	26.6 π
1000	16.7	18.8	33.4 π
1200	20.0	22.6	40 π

Responses to Reviewer #3

Reviewer #3: The article "Spatiotemporal control of upconversion photochromic evolution from a single emitter through interfacial energy transfer" by Long Yan et al. presents an approach suitable to enhance upconversion luminescence intensity and 'switch' red/green colour by pulse width modulation or increasing pump power intensity of CW photo-excitation.

Although the article is well written, the samples are well prepared and characterized, the figures are clear and interesting, the discussion is reasonable I am not convinced about high / sufficient novelty of the presented studies, which instead I consider as reshaping the already well known effects. Not only the pulse width modulation on the UC colour has been shown before (ref.7 and other not cited here for YbHo, but also YbEr), but spatiotemporal colour 'splitting' was also demonstrated. Moreover, I do not

think the discussion provides an 'in-depth insight into the photophysics', but it is rather mechanistic and qualitative description. The UC intensity growth has been also demonstrated by increased sensitizer content as well as by surface passivation with the undoped shell.

Overall, I am of the opinion the manuscript lacks clear and strong novelty.

Reply: We are grateful to this reviewer for the critical comments. This motivates us to further and deeply think about the importance and novelty of our work, which are useful and valuable to further improve the quality of our work, and to gain an in-depth insight into the photophysics. However, with all due respect, we feel that the reviewer might have misunderstood the novelty and importance of our work, perhaps because of our unclear writing in the manuscript. Here we would like to justify the novelty and potential impact of our work within the context of the three comments underlined above.

(1) The statement of “consider as reshaping the already well known effects”, which is based on the viewpoint of “Not only the pulse width modulation on the UC colour has been shown before (ref.7 and other not cited here for YbHo, but also YbEr), but spatiotemporal colour 'splitting' was also demonstrated.” We feel that the reviewer might have misunderstood the key achievements of our work. We agree that some reports on pulse width modulation or spatiotemporal colour 'splitting' of upconversion have been published over the past years, see refs 7,8,13,16, 27-29,39 in the manuscript. However, there are still many disadvantages and shortcomings for these reports. For the steady-state upconversion for the colour tuning, e.g., by designing a multi-layer core-shell (MLCS) nanostructure with two different excitation wavelengths (e.g., see ref 26: *ACS Nano*, **2017**, 11, 3289-3297; ref 30: *Adv. Mater.*, **2018**, 30, e1801726), which impose a critical requirement for the synthesis of high-quality MLCS structure, and also have a complex excitation system because of using two lasers. While for non-steady-state upconversion, one effective way is designing cross relaxation by introducing another lanthanide ion which could concurrently result in a decline in luminescence intensity, such as the Yb/Ho/Ce systems (e.g., see ref 8 (named ref 7 in the original version): *Nat. Nanotech.*, **2015**, 10, 237-242; ref 40: *Nat. Commun.*, **2022**, 13, 4741). Besides, the research on spatiotemporal colour 'splitting' focused on dual-emitting centers with different decay times such as Tm³⁺ and Tb³⁺, Tm³⁺ and Mn²⁺ (see ref 26: *ACS Nano*, **2017**, 11, 3289–3297; ref 5: *Nat. Commun.*, **2017**, 8, 899.) which led to complex ionic interactions and less efficient upconversion. For the single emitter, it is more difficult to modulate both intensity and lifetime to achieve spatiotemporal colour 'splitting'. The temporally reversible green-red emission color of Er³⁺ was recently reported (see ref 27: *Angew. Chem. Int. Ed.*, **2022**, 61, e202212089.), however it depends sensitively on the sample size due to energy migration among sensitizers, which only works well in micron-meter size particles instead of nanoscale particles. Also, it should be noted that most of those works seldom report the upconversion quantum yield, which is one of key factors to quantitatively describe the upconversion performance. Moreover, search the new strategy for enhancement of upconversion would break the bottleneck of how to

further enhance the intensity when the doping concentration promotes to 100%. which is a generally existed problem for conventional luminescent materials. Therefore, it still challenging to realize the spatiotemporal control of upconversion with fine and tunable emission colours from a single emitter.

Aiming to overcome the above bottlenecks, here in the present work we demonstrate a new strategy to realize the spatiotemporal control of upconversion from a single emitter through constructing the interfacial energy transfer (IET) channel. In particular, we obtained the following novel and important results:

- (i) We designed a core-shell-shell structure to greatly enhance the upconversion of Er sublattice by outside coating a sensitizing Yb shell through IET. Based on this design, the green upconversion shows an enhancement with two orders of magnitude, and the red upconversion shows an enhancement with one order of magnitude. Moreover, the upconversion quantum yield was also increased by comparison to the control sample without the Yb shell.
- (ii) We demonstrated a new strategy to enhance upconversion by selectively manipulating the Er-Yb ionic interactions at core-shell interfacial region. We confirmed that the backward energy transfer is a leading factor to limit upconversion, which has a much more sensitive dependence on the ionic separation. And we further obtained the critical distance R_c at which the spontaneous emission rate is equal to the energy transfer rate. R_c is no more than 0.6 nm for backward energy transfer much smaller than forward energy transfer (~2.8 nm) for Er-Yb couple. This is the first quantitative investigation on this issue for UCNPs.
- (iii) We further demonstrated the green/red dual-colour switchable output by precise control of the rise and decay dynamics through the use of short pulse laser excitation and time-gating technique. We show that a properly spatial separation of Er and Yb sublattices resulted in a temporal control of upconversion dynamics. Our design shows a faster rise time for green emission than red emission as an efficient IET, which differs greatly from the common $\text{NaErF}_4@ \text{NaYF}_4$ and $\text{NaErF}_4:\text{Ho}@ \text{NaYF}_4$ control samples that is not able to present a colour change. Moreover, this design also shows different decay times allowing to make a colour tuning by using the time-gating technique. Such temporal control of upconversion permits more functions in addition to the steady-state control of upconversion in a single emitter.
- (iv) The smart spatiotemporal control of emission colours in nanoparticles permits great chances for frontier photonic applications. And we demonstrated their frontier applications in information storage and security, multi-level anticounterfeiting, and speed monitoring. In particular, the speed sensitivity is as high as $1.70 (\text{km/h})^{-1}$, higher than the recent reports. These suggest the great potential of our design for broad photonic applications.

Therefore our results are interesting and important and indeed not “consider as reshaping the already well known effects”.

(2) **The statement of “do not provide an 'in-depth insight into the photophysics”**. In the original version, we attempted to present an "in-depth insight into the photophysics" by the careful and systematic investigations on the content of Yb in the sensitizing shell, its thickness, spatial control over upconversion dynamics between Er and Yb sublattices on the nanometer length scale, together with series of ways to tune color-switchable output. We agree that these investigations somewhat looks like "mechanistic and qualitative description". However, we should note that there are important and interesting phenomena were observed, and many of them were observed for the first time, such as enhancing upconversion by spatial control of ionic interactions between activator (Er) and sensitizer (Yb) sublattices, achieving the temporal tuning of emission colors in these samples which are non-available for conventional NaYF₄:Yb/Er@NaYF₄ UCNPs. On the other hand, this statement also motivated us to deeply consider the scientific importance of our work, and in particular how to present more quantitative instead of qualitative results and discussions. For this aim, we further carried out a series of new measurements and discussions. In particular, we measured the upconversion quantum yield (QY) for our design, and we confirmed that QY is indeed higher than that of the control sample without Yb shell (Fig. 2a-d; Supplementary Fig. 5). And both upconversion intensity and QY can be further enhanced by a spatial control of interfacial interactions (Fig. 3cd; Supplementary Fig. 18). We further calculated the forward and backward energy transfer rate, and obtained the critical Er-Yb distance R_c at which the spontaneous emission rate is equal to the energy transfer rate. And we found that R_c for backward energy transfer (≤ 0.6 nm) is much smaller than that of forward energy transfer (~ 2.8 nm) (Fig. 3f; Supplementary Fig. 20). This is the first time for a quantitative description on the IET for UCNPs.

(3) **The statement of “the UC intensity growth has been also demonstrated by increased sensitizer content as well as by surface passivation”**. We agree that increasing sensitizer content and surface passivation are two commonly used ways to enhance upconversion. However, it should be noted that our design does not aims to enhance the upconversion by merely increasing Yb content. Alternatively, we aim to demonstrate our design by using a sensitizing shell outside the activator core, which can break up the intrinsic constraint on doping concentration in conventional codoping systems which is smaller than 100% because the existence of activator. Moreover, for activator lattice (namely 100% doping) there is no site was left for sensitizer. So, how to realize and enhance the upconversion of activator lattice is always a question. And here we confirmed that our design with spatial separation of Er and Yb sublattices in nanostructure is a facile but effective model for enhancing upconversion and tuning its emission colours. On the other hand, the increased sensitizer content is used to prove that the Yb shell instead of Yb-doped shell has the best performance for enhancing upconversion; it is not a simple content increase as it may seen. Regarding the surface passivation by an inert shell is a regular method for UCNPs and was widely employed in this field. And we did not highlight its novelty but a normal design.

In addition to the above, we also found the following important results: i) The energy migration would depress its energy utilization and limit the upconversion for Yb sensitizing shell (Fig. 2f); ii) The upconversion enhancement factor is markedly higher at lower pump power densities (Supplementary Fig. 13), which is important for bioapplications; iii) The upconversion emission intensity of this sample is much higher than that of the commonly reported NaYF₄:Yb/Er(20/2 mol%)@NaYF₄ core-shell control nanoparticles, in particular at the lower pump power densities (e.g., it is more than one order of magnitude when the excitation is below 2 W/cm²), see Fig. 3e; Supplementary Fig. 19. iii) We have further made a careful and thorough revision according to the comments and constructive suggestions of all reviewers, see the red highlight sentences and newly added/updated Supplementary Figures 1-3,5,8-9,13,15-16,18-20,22,24,26,31-32, Table S5,S6 and experimental details in the revised *Supplementary Information*.

Therefore, we believe that our systematic work presents a substantial breakthrough and a conceptual advance in the field of luminescent materials instead of "lacks clear and strong novelty" assessed by the reviewer #3. The discovery of spatiotemporal control of activator (Er) and sensitizer (Yb) sublattices provides a new visual angle to improve the understanding of the ionic interactions, and break the stereotypes roles about them towards much more efficient and versatile manipulation of the photon upconversion, thus providing an entirely new level of knowledge with significant implications for frontier applications. We certainly hope the reviewer concurs. To make a more clear clarification, we have rewritten the maintext properly, and added series of new experimental results as well as discussions. And now we hope the reviewer will perhaps find our work more suitable for publication in *Nature Communications*.

In the revised version, the following sentences were added:

(pp 8): "Another merit lies in that the upconversion enhancement factor is markedly higher at lower pump power densities (Supplementary Fig. 13), which is important for bioapplications."

(pp 9): "This is also supported by the prolonged lifetimes of red and green emissions (Supplementary Fig. 17) and increased quantum yield (Supplementary Fig. 18). Here it is worth noting that the upconversion of this sample is much higher than that of the commonly reported NaYF₄:Yb/Er(20/2 mol%)@NaYF₄ core-shell control nanoparticles, in particular at the lower pump power densities (Fig. 3e; Supplementary Fig. 19). "

(pp 9-10): "We further calculated the critical distance (R_c) at which the energy transfer rate is equal to the emission rate for a donor ion (Fig. 3f; Supplementary Fig. 20)⁴⁹⁻⁵¹. It is no more than 0.6 nm for backward energy transfer, much shorter than the forward energy transfer (~2.8 nm). "

(pp 15): "..., and the quantum yield has aslo been raised. A fine manipulation of forward and backward energy transfer by employing a thickness-tunable interlayer further contributes to the upconversion emission. "

The updated Fig. 3e,f:

Fig. 3 e A comparison of upconversion emission intensity from (c) sample with 2.1 nm thick NaYbF₄ interlayer (ErHo@Y@Yb@Y) and control NaYF₄:Yb/Er(20/2 mol)@NaYF₄ core-shell nanoparticles (YbEr@Y) under identical measurement condition. **f** Dependence of energy transfer rate between Yb³⁺ and Er³⁺ on the ionic distance.

The newly added Supplementary Figs. 5,13,18,19:

Supplementary Fig. 5. Upconversion quantum yield from NaErF₄:Ho(0.5 mol%)@NaYbF₄@NaYF₄ (ErHo@Yb@Y) core-shell-shell and NaErF₄:Ho(0.5 mol%)@NaYF₄ (ErHo@Y) core-shell nanoparticles under 980 nm excitation with variable power densities.

Supplementary Fig. 13. (a,b) Upconversion emission spectra of NaErF₄:Ho(0.5 mol%)@NaYbF₄@NaYF₄ core-shell-shell (ErHo@Yb@Y) and NaErF₄:Ho(0.5 mol%)@NaYF₄ core-shell (ErHo@Y) nanoparticles under 980 nm excitation at different pump power densities. (c,d) Dependence of the upconversion intensity of green (c) and red (d) emissions from (a,b) samples.

Supplementary Fig. 18. Upconversion quantum yield obtained from NaErF₄:Ho(0.5 mol%)@NaYF₄@NaYbF₄@NaYF₄ (ErHo@Y@Yb@Y) core-multishell and NaErF₄:Ho(0.5 mol%)@NaYbF₄@NaYF₄ (ErHo@Yb@Y) core-shell-shell nanoparticles under 980 nm excitation with variable power densities.

Supplementary Fig. 19. (a,b) Upconversion emission spectra of NaErF₄:Ho(0.5 mol%)@NaYF₄@NaYbF₄ @NaYF₄ core-multishell (ErHo@Y@Yb@Y) and NaYF₄:Yb/Er(20/2 mol%)@NaYF₄ core-shell (Yb20Er2@Y) nanoparticles under 980 nm excitation at different pump power densities. (c,d) Dependence of the upconversion intensity of green (c) and red (d) emissions from (a,b) samples.

Responses to Reviewer #4

Reviewer #4: In this manuscript, the authors demonstrate bright visible light emission and color tuning from the NaErF₄:Ho(0.5%)@NaYF₄:Yb@NaYF₄ core-shell-shell UCNPs via interfacial energy transfer process. By changing Yb concentration in the NaYF₄:Yb layer and the thickness of the NaYF₄:Yb layer, highly luminescent UCNPs could be obtained. The modulation of the pulse width and the power of the incident laser enabled the fine tuning of the emission color from the core-shell-shell UCNPs and green to red color change was shown. By utilizing these unique properties, anti-counterfeiting patterns were successfully demonstrated. Also, the authors converted temporal photochromic evolution to spatial color distribution and they utilized this for speed monitoring. These results are quite interesting and the reviewer thinks that this study is interesting to the broad readership in Nature Communications. However, there are many flaws which should be addressed. In some parts, figure captions are not consistent with figures and the figure is not consistent with explanation in the main text as follows. These should be carefully checked, and following issues should be addressed before acceptance.

Reply: Thanks for the Reviewer's positive comments and valuable suggestions, and we have carefully checked and revised the manuscript following them.

Q1. While Fig. 2e shows slope values dependent on Yb³⁺ concentrations in NaYF₄:Yb layer, Fig. 2e caption described the dependence of lifetime values of Er³⁺ emissions on Yb³⁺ concentration. Fig. 2e caption seems to be wrong.

Reply: Thanks for the careful reading. In the Fig. 2e caption, the y axis should be "slope values" instead of "lifetime". Sorry for this mistake and we have corrected it in revised manuscript. This can also refer to the response to the comment Q4 of reviewer #2.

Q2. In page 6, it is described that the presence of Yb³⁺ can further improve the green emission and consequently promote the CR. How did the presence of Yb³⁺ in the shell improve green emission from the Er³⁺ in the core and promote the CR? According to the previous study, CR in Er³⁺ increases when the Er³⁺ concentration increases in the core (Angew. Chem. Int. Ed. 2017, 2017, 56, 7605 and Nanoscale, 2017, 9, 7941). If the concentration of Yb³⁺ increases in the NaErF₄:Ho(0.5%) core, the presence of Yb³⁺ would result in the decrease of Er³⁺ amounts and increase the distance between Er³⁺ ions in the core. Then, CR ($[(^2H_{11/2}, ^4S_{3/2}); ^4I_{9/2}] \rightarrow [^4F_{9/2}; ^4F_{9/2}]$) will decrease and green emission will be improved. However, in this study, Yb³⁺ is present in the shell and thus, related explanation need to be provided. In addition, if the presence of Yb³⁺ can further improve the green emission, it will suppress the CR. As the authors explained in page 6, stronger red emission than green emission is due to CR among Er³⁺ ions. Thus, the sentence "the presence of Yb³⁺ can further improve the green emission and consequently promote the CR" does not seem to make sense.

Reply: Thanks for the comment. There exists intrinsic CR between Er³⁺ ions in NaErF₄ matrix that can promote the population of red emitting level through the CR of $[(^2H_{11/2}, ^4S_{3/2}); ^4I_{9/2}] \rightarrow [^4F_{9/2}; ^4F_{9/2}]$ (e.g., see ref 44: Nanoscale, 2018, 10, 17949-17957). This is an additional population way apart from the regular ETU process. In this study, the presence of Yb³⁺ in the interlayer increases the absorption at 980 nm by designing the NaYbF₄ shell and greatly enhances the upconversion of NaErF₄ through interfacial energy transfer from Yb³⁺ to Er³⁺, see Fig. 2c. The enhancement of green emission means that more Er³⁺ ions were populated at this energy level, and accordingly the CR occurring at this level may further take place and contribute to the population of double Er³⁺ ions to its red-emitting level. This additional process would reduce the slope value of the red upconversion (Fig. 2e).

For a clearer description, we revised the sentence in the original version, and now it is shown as " The presence of Yb³⁺ in the interlayer can further improve the population of Er³⁺ at its green emitting levels through IET and consequently promote the cross relaxation of $[(^2H_{11/2}, ^4S_{3/2}); ^4I_{9/2}] \rightarrow [^4F_{9/2}; ^4F_{9/2}]$. This adds a new channel to increase the population of red emitting ⁴F_{9/2} level and reduce the slope value." in the revised version (pp 6-7). This can also refer to the response to comment Q5 of reviewer #2.

Q3. The authors explained that the presence of Yb³⁺ can further improve the green

emission and consequently promote the CR, and this provides a new channel to help the population of the red emitting level and reduce the slope value. If there is a new channel affects the PL intensity, decay and/or rise behavior might change. Thus, the results can be explained with time-resolved PL spectra. In Fig. 2c, PL intensity of the NaErF₄:Ho(5%)@NaYF₄:Yb(x)@NaYF₄ core-shell-shell UCNPs increased with increasing Yb³⁺ concentration. However, decay time of the core-shell-shell UCNPs showed increased and then decreased as the Yb concentration increased in Fig. S11 and Table S1. Why did the decay time show different trend from the PL intensity change with varying Yb concentration?

Author Reply: In NaErF₄:Ho(0.5 mol%)@NaYF₄:Yb(0-100 mol%)@NaYF₄ core-shell-shell UCNPs, the presence of Yb³⁺ could improve the upconversion intensity through increasing absorption of 980 nm laser energy and following efficient IET, and the upconversion was enhanced (see Fig. 2c). As shown in Supplementary Fig. 28b, it can be observed that the sample with NaYbF₄ interlayer has a faster rise time than that without NaYbF₄ interlayer. At low Yb³⁺ doping concentration, the IET from Yb³⁺ to Er³⁺ dominates the ionic interactions resulting in enhancement of both intensity and lifetime. However, much higher Yb³⁺ concentration would begin to cause back energy transfer from Er³⁺ to Yb³⁺, which would suppress and even reduce the intensity and lifetime. Indeed, the upconversion intensity shows a saturation trend when Yb³⁺ is higher than 20 mol%, see Fig. 3a. This confirmed the occurrence of back energy transfer, and caused the decrease of lifetime at higher Yb³⁺ concentration (Table S1). We also calculated the critical distance (R_c) at which energy transfer rate is equal to emission rate of sensitizer, and we found that R_c is no more than 0.6 nm for back energy transfer, much smaller than the forward energy transfer (~2.8 nm). This presents a deeper proof for the investigation of ionic interactions at the core-shell interface.

For a clearer understanding, we added the following sentences in the revised version:

(pp 9): " Considering the energy level diagram of Er³⁺ and Yb³⁺, back energy transfer may occur for the green and red emitting energy levels of Er³⁺ (Supplementary Fig. 14a)."

(pp 9-10): " We further calculated the critical distance (R_c) at which the energy transfer rate is equal to the emission rate for a donor ion (Fig. 3f; Supplementary Fig. 20)⁴⁹⁻⁵¹. It is no more than 0.6 nm for backward energy transfer, much shorter than the forward energy transfer (~2.8 nm). "

The newly added Fig. 3f and Supplementary Fig. 20:

Fig. 3f Dependence of energy transfer rate between Yb^{3+} and Er^{3+} on the ionic distance.

Supplementary Fig. 20. (a-c) Decay curves of Er^{3+} at its (a) $^4\text{S}_{3/2}$ (550 nm) and (b) $^4\text{F}_{9/2}$ (660 nm) from $\text{NaErF}_4:\text{Ho}(0.5 \text{ mol}\%)\text{@NaYF}_4\text{@NaYbF}_4\text{@NaYF}_4$ core-multishell nanoparticles with increasing NaYF_4 interlayer thickness (0-8.7 nm) and that from the control sample of $\text{NaYF}_4\text{@NaYbF}_4\text{@NaYF}_4$ core-shell-shell nanoparticles. (c) Decay curves of Yb^{3+} at its $^2\text{F}_{5/2}$ (1020 nm) in $\text{NaErF}_4:\text{Ho}(0.5 \text{ mol}\%)\text{@NaYF}_4\text{@NaYbF}_4\text{@NaYF}_4$ core-multishell nanoparticles with increasing

NaYF₄ interlayer thickness (0-8.7 nm) and that from the control sample of NaYF₄@NaYbF₄@NaYF₄ core-shell-shell nanoparticles.

Q4. When the Fig. 3d and 3e, NaYF₄ interlayer thickness values are not consistent. Moreover, the PL spectra data (Fig. 3d) are not consistent with normalized intensity (Fig. 3e) of NaErF₄:Ho(5%)@NaYF₄@NaYbF₄@NaYF₄ UCNPs with varying NaYF₄ interlayer thickness. The figure and data should be clear.

Reply: Thanks for the careful reading. There was a mistake in the plotting of the data (it was Fig. 2f by mistake). In the revised version, we corrected it by using the correct data. The data in Fig. 3e are correct. Sorry for this mistake. This can also refer to the response to comment Q7 of reviewer #2.

Q5. In this study, NaErF₄:Ho(5%)@NaYF₄@NaYbF₄@NaYF₄ with NaYF₄ interlayer thickness of 2.1 nm showed the highest PL intensity. However, the authors did not investigate the color tuning for the NaErF₄:Ho(5%)@NaYF₄(2.1 nm)@NaYbF₄@NaYF₄ UCNPs. Also, the authors did not use the NaErF₄:Ho(5%)@NaYF₄(2.1 nm)@NaYbF₄@NaYF₄ UCNPs for optical applications. Why?

Reply: Thanks for your comment. The investigations on the color tuning for the NaErF₄:Ho(0.5%)@NaYF₄(2.1 nm)@NaYbF₄@NaYF₄ nanoparticles were shown in Figure S16d from the original SI (renamed as Supplementary Figs. 25d,26d in revised SI). And the investigations on the color tuning for the NaErF₄:Ho(0.5%)@NaYF₄(2.1 nm)@NaYbF₄@NaYF₄ nanoparticles were shown in Supplementary Figs. 21f,22f in revised SI). Although the NaErF₄:Ho(0.5%)@NaYF₄(2.1 nm)@NaYbF₄@NaYF₄ nanoparticle showed the highest PL intensity, its performance of dynamic control on emission color was not as good as the NaErF₄:Ho(0.5% mol)@NaYF₄@NaYF₄:Yb(20% mol)@NaYF₄ as shown in the newly added Supplementary Fig. 32 (plotted from Supplementary Figs. 21f and 25d). The latter sample showed larger color gamut from green to red color. Thus, we selected the latter sample for application.

The newly added Supplementary Fig. 32:

Supplementary Fig. 32. Dependence of red/green emission intensity (R/G) ratio on

pulse widths and c.w. of 980 nm excitation laser for NaErF₄:Ho(0.5%)@NaYF₄(2.1 nm)@NaYbF₄@NaYF₄ and NaErF₄:Ho(0.5%)@NaYF₄:Yb(20%)@NaYF₄ core-multishell nanoparticles.

Q6. The NaErF₄:Ho(5%)@NaYF₄@NaYbF₄@NaYF₄ UCNPs showed PL enhancement and then decrease of the PL intensity with increasing the NaYF₄ interlayer thickness. According to the authors, it is due to effective depression of energy loss due to backward energy transfer channels. The presence of NaYF₄ interlayer between NaErF₄:Ho and NaYbF₄ may depress energy transfer from Yb³⁺ to Er³⁺ and backward energy transfer from Er³⁺ in the core to Yb³⁺ in the NaYbF₄ shell. More explanation as to why the backward energy transfer was more effectively suppressed needs to be provided.

Reply: There is a small energy mismatch for the Yb³⁺ transition (²F_{7/2}→²F_{5/2}) and that of Er³⁺ transitions: ⁴F_{7/2}→⁴I_{11/2} and ⁴F_{9/2}→⁴I_{13/2}, as shown in the newly added Supplementary Fig. 14a in revised SI. While for the forward energy transfer from Yb³⁺ (²F_{5/2}) to Er³⁺ (⁴I_{11/2}) it is resonant. This means that the forward and backward energy transfer may have different separation-dependent features. Namely, the back energy transfer should only occur in a much short separation due to the small energy mismatch. Just because there are strong backward energy transfer at much smaller NaYF₄ interlayer thickness, the upconversion can be enhanced by introducing a suitable NaYF₄ interlayer that can greatly reduce the backward energy transfer but has limited impact on forward energy transfer. Thus, the optimal upconversion was observed in the sample with 2.1 nm NaYF₄. However, further increasing NaYF₄ interlayer thickness, the forward energy transfer also becomes weak and the upconversion resulted in a decline.

In order to further explain this, here we calculated the critical separation R_c , at which the spontaneous emission rate is equal to the energy transfer rate, for forward and backward energy transfers. The results are shown in Fig. 3f and Supplementary Fig. 20. The R_c for forward energy transfer is 2.8 nm, while for back energy transfer, it is no more than 0.6 nm (for green emitting level; for the red emitting level it is smaller than 0.5 nm). This further explains why a 2.1 nm NaYF₄ interlayer shows the highest upconversion intensity. That thickness is a balance between the helpful forward and harmful backward energy transfer towards the highest emission intensity.

In the revised manuscript (pp 9-10), we added the following sentences for a quantitative comparison: "We further calculated the critical distance (R_c) at which the energy transfer rate is equal to the emission rate for a donor ion (Fig. 3f; Supplementary Fig. 20) ⁴⁹⁻⁵¹. It is no more than 0.6 nm for backward energy transfer, much shorter than the forward energy transfer (~2.8 nm)."

Supplementary Fig. 14a. Schematic of possible back energy transfer (BET) processes from Er^{3+} ($^4\text{F}_{7/2}$, $^4\text{F}_{9/2}$ and $^4\text{I}_{11/2}$) to Yb^{3+} . There is a small mismatch for BET from $^4\text{F}_{7/2}$ and $^4\text{F}_{9/2}$ states of Er^{3+} , while it is resonant for that from $^4\text{I}_{11/2}$ state of Er^{3+} .

Q7. In Fig. 3, green and red emission intensities increased when NaYF_4 interlayer thickness reached 2.1 nm and then it decreased when the NaYF_4 interlayer thickness was larger than 2.1 nm. The decay time measured for green emission showed the same trend. However, decay time measured for red emission increased when the NaYF_4 interlayer thickness reached 4.3 nm and it was shortened when the NaYF_4 interlayer thickness was 8.7 nm. Unlike the green emission, for red emission, why is the trend in the PL intensity change different from that in the decay time change?

Reply: In our study, the enhanced upconversion of Er^{3+} from the sample with a 2.1-nm thick NaYF_4 interlayer is due to a balance between the forward and backward energy transfer which show different dependence on ionic separation. When the NaYF_4 inter layer thickness was larger than 2.1 nm, the IET upconversion channel from Yb^{3+} to Er^{3+} became weakened leading to a decrease of intensity and lifetime. However, the effect of existing CR process of $[(^2\text{H}_{11/2}, ^4\text{S}_{3/2}); ^4\text{I}_{9/2}] \rightarrow [^4\text{F}_{9/2}; ^4\text{F}_{9/2}]$ on upconversion process became remarkable and led to a further extension of the lifetime of the red-emitting $^4\text{F}_{9/2}$ level (see Supplementary Fig. 17).

Q8. Fig. S12(b-f) shows TEM images of $\text{NaErF}_4:\text{Ho}(5\%)@\text{NaYF}_4$ core-shell UCNPs with increased thickness of NaYF_4 layer. According to the labeling, it is expected that the sizes of core-shell UCNPs correspond to 15.3, 16.5, 18.5, 22.9, and 31.1 nm for (b), (c), (d), (e), and (f), respectively. However, the sizes of the core-shell UCNPs in (a), (c), and (e) look very similar. In addition, brightness contrast between core and shell regions is expected to be shown due to large difference of atomic numbers of Er in the core and Y in the shell. However, the brightness contrast is only observed in Fig. S12f. Is it due to thin shell thickness for Fig. S12(b-e)?

Reply: Thanks for the comment. We apologize for the wrong sequence number and the correct order given in Fig. S12 in the original version. Here we further supplemented the TEM images for the sample after coating each shell layer, and added them as Supplementary Fig. 15 in revised SI. These TEM images present a clear comparison in both size and morphology. The size of these nanoparticles was counted by measuring the length in TEM graph using Digital Micrograph. The size distribution was a range of all particle statistics and the particle size showed the average result. When the NaYF_4 layer is thin, the size of nanoparticles have a small

change (see Fig. S12g), which may lead to a similar impression (Fig. S12a-c). And the contrast is not obvious because of the thin shell. When the shell became thicker (Fig. R8d), the contrast due to large difference of atomic numbers begins to be observed clearly (Fig. S12f).

Figure S12 in the original version. TEM images of (a) NaErF₄:Ho(0.5 mol%) core and (b-f) NaErF₄:Ho(0.5 mol%)@NaYF₄ core-shell nanoparticles with the increased thickness of NaYF₄ layer. (g) The corresponding particle size distributions of (a-f) samples.

The updated Supplementary Fig. 15:

Supplementary Fig. 15. (a-d) TEM images of NaErF₄:Ho(0.5 mol%)@NaYF₄@NaYbF₄@NaYF₄ core-multishell nanoparticles with increasing NaYbF₄ layer thickness at each stage. C, CS, CSS and CSSS represents the NaErF₄:Ho(0.5 mol%) core, NaErF₄:Ho(0.5 mol%)@NaYF₄ core-shell, NaErF₄:Ho(0.5 mol%)@NaYF₄@NaYbF₄ core-shell-shell and NaErF₄:Ho(0.5 mol%)@NaYF₄@NaYbF₄@NaYF₄ core-shell-shell-shell nanoparticles.

mol%>@NaYF₄@NaYbF₄@NaYF₄ core-multishell nanoparticles with different NaYF₄ inter layer thicknesses (0.5-8.7 nm), respectively. (e) Corresponding particle size of the NaErF₄:Ho(0.5 mol%) core and coated with different NaYF₄ interlayer thicknesses.

Q9. In page 5, $^4S_{2/3} \rightarrow ^4I_{15/2}$ should be $^4S_{3/2} \rightarrow ^4I_{15/2}$. In page 11, red-emitting $^4F_{7/2}$ should be $^4F_{9/2}$.

Reply: Thanks for your careful review, and we have corrected them in the revised version.

Q10. In page 11, it is described that the population of Er³⁺ at the red emitting $^4F_{9/2}$ level can be improved with a decline of the intensity-power slope values (Fig. 2f). However, Fig. 2f shows PL spectra of NaErF₄:Ho(0.5%)@NaYbF₄@NaYF₄ UCNPs with varying NaYbF₄ thickness. How did the Fig. 2f show a decline of the intensity-power slope values?

Reply: Thanks for your careful review. The “Fig.2f” here should be “Fig. 2e”, and we have corrected it in the revised version.

Q11. The authors claimed that emission intensity was enhanced by IET from NaYbF₄ to NaErF₄:Ho(0.5%). Then, what are upconversion quantum yields (UCQYs) of NaErF₄:Ho(5%)@NaYbF₄@NaYF₄ UCNPs. Is there a power density dependence of UCQYs?

Reply: We have measured the UCQYs of NaErF₄:Ho(0.5%)@NaYbF₄@NaYF₄ and NaErF₄:Ho(0.5%)@NaYF₄ nanoparticles, see the newly added Supplementary Fig. 5, showing an increase with power density. And the highest UCQY is 0.47% at a pump power density of 103.93 W cm⁻². As a control, we also measured the UCQYs of NaErF₄:Ho(0.5%)@NaYF₄ nanoparticles, which also showed an increasing trend with elevating pump power density, but much lower than the sample at all power densities. This further proved the effectiveness our design. These data were added as Supplementary Fig. 5 and the QY measurement method was added in the characterization section in the revised SI, and in the maintext (pp 6) the following sentence was added: " More importantly, the upconversion quantum yield also shows an increase by comparison to the control sample without NaYbF₄ layer (Supplementary Fig. 5)."

The newly added Supplementary Fig. 5:

Supplementary Fig. 5. Upconversion quantum yield from NaErF₄:Ho(0.5 mol%)@NaYbF₄@NaYF₄ (ErHo@Yb@Y) core-shell-shell and NaErF₄:Ho(0.5 mol%)@NaYF₄ (ErHo@Y) core-shell nanoparticles under 980 nm excitation with variable power densities.

Q12. In Fig. 5 caption, NaErF₄(0.5mol%)@NaYF₄:Yb(20 mol%)@NaYF₄ UCNPs seems to be NaErF₄:Ho(0.5 mol%)@NaYF₄:Yb(20 mol%)@NaYF₄. It should be checked. Why did the authors used NaErF₄:Ho(0.5 mol%)@NaYF₄:Yb(20 mol%)@NaYF₄ UCNPs for applications although NaErF₄:Ho(0.5 mol%)@NaYF₄:Yb(10 mol%)@NaYF₄ UCNPs showed higher G/R ratio than NaErF₄:Ho(0.5 mol%)@NaYF₄:Yb(20 mol%)@NaYF₄ UCNPs?

Reply: Thanks for your careful reading and comment. In Fig. 5 caption, it indeed should be NaErF₄:Ho(0.5 mol%)@NaYF₄:Yb(20 mol%)@NaYF₄, and it has been corrected in the revised manuscript. And the reasons of using NaErF₄:Ho(0.5 mol%)@NaYF₄:Yb(20 mol%)@NaYF₄ UCNPs for applications are because of their higher brightness, together with wider color change gamut from green to red colors under temporal modulation (see Supplementary Fig. 32). This can also refer to the response to the comment Q5 of this reviewer.

In the revised version, we have corrected the "NaErF₄(0.5 mol%)@NaYF₄:Yb(20 mol%)@NaYF₄" in Fig. 5 caption: "f Speed-dependent upconversion emission spectra of NaErF₄:Ho(0.5 mol%)@NaYF₄:Yb(20 mol%)@NaYF₄ core-shell-shell nanoparticles under steady-state 980 nm laser excitation".

Q13. The authors claim that the maximum speed sensitivity was much higher than the recent reports. However, in ref. 49, the highest Sa value is 0.045 s/cm. On the other hand, in this study, Sa is 1.29 (km/h)⁻¹, which is the same as 0.0469 s/cm. Thus, Sa (1.29 h/km) in this study is not much higher than the previous study.

Reply: We have revised the accurate expression in this section. In the original manuscript, we chose the LIR of 654 nm/540 nm by using their peak intensity, which would cause somewhat error. Here we employed the integrated spectral intensity for LIR values and the green emission covers the two peaks (540 nm and 521 nm). The results is shown in Fig. 5g,h. It can be found that the calculated S_a value is 1.70 (km/h)⁻¹, equal to 0.0612 s/cm, which is higher than the previous study. In the revised

version, we have updated the figure 5g,h and S_a value in pp 14, "and the maximum speed sensitivity was further calculated to be 1.70 (km/h)^{-1} ".

The updated Fig. 5g,h:

Fig. 5. g Plot of luminescence intensity ratio (LIR) versus speed together with the fitting by an exponential function. **h** Dependence of calculated absolute (S_a) and relative (S_r) sensitivity versus speed.

Q14. In this study, very interesting results were shown in Fig. 5. However, related explanation is not enough. For example, an upper left image (red square) in Fig. 5b showed color change from red for long pulse excitation to mixed color of red and green in QR code for short pulse excitation. That is, in some regions, red color changed to green color and in other regions, red color was maintained. Then, were different compositions of the UCNPs patterned together for the image shown in Fig. 5b? Since the authors showed that core-shell-shell UCNPs showed green and red emission for short and long pulse excitations, respectively, mixed color of red and green shown in the upper right image of Fig. 5b needs to be explained in more detail.

Reply: In Fig. 5b, the pattern was consisted of QR code and square pattern painting with $\text{NaErF}_4:\text{Ho}(0.5 \text{ mol}\%)\text{@NaYF}_4:\text{Yb}(20 \text{ mol}\%)\text{@NaYF}_4$ and $\text{NaErF}_4:\text{Ho}(0.5 \text{ mol}\%)\text{@NaYF}_4:\text{Yb}(10 \text{ mol}\%)$ respectively as shown in Fig. 5b in manuscript. With long pulse excitation, the pattern showed the red square (i). Under excitation with increased power density (ii) or short pulse (iii), the QR code could be decoded due to the color change from red to green. With 600 nm short-pass filter under same condition(iv-vi), only the green emission regions could be identified. The iv-vi respectively corresponded to i-iii. These were clearly added in the Fig. 5b caption. The details of this application were added in the "Supplementary Methods" and the color-tuning property of the used nanoparticles was added in Supplementary Table 5 in SI. And the Fig. 5b caption was rewritten as " Decoding the pattern information (top left panel) through high pump power (top middle panel) or short pulse irradiation (top right panel). The bottom panels show the photographs taken after a 600 nm short-pass filter. Scale bar, 5 mm." in the revised manuscript to replace the original version.

Fig. 5b in the manuscript.

Supplementary Table 5. A summary of the color-tuning properties of the samples used in the preparation of QR pattern.

Samples	Emission colors		
	Long width	Short width	High power
NaYF ₄ :Yb/Er(20/2 mol%)@NaYF ₄	Green	Green	Green
NaErF ₄ :Ho(0.5 mol%)@NaYF ₄ :Yb(20 mol%)@NaYF ₄	Red	Green	Green
NaErF ₄ :Ho(0.5 mol%)@NaYF ₄	Red	Red	Red
NaYbF ₄ :Er(1 mol%)@NaYF ₄	Green	Green	Red

REVIEWER COMMENTS

Reviewer #1 (Remarks to the Author):

The authors have adequately responded to my concerns.

Reviewer #3 (Remarks to the Author):

1. That is good the UCQY measurements were added, but more detailed information and exemplary data are requested to demonstrate how UCQY were measured and analyzed. These are not trivial measurements. How pump power dependent UCQY were obtained (by changing LD current?), especially if $QY < 1\%$?
2. What do the variations in XRD patterns (e.g. Fig.S9) mean from crystallographic perspective ?
3. Ho^{3+} energy levels are much more complex than just 5I6 and actually some levels exist, which overlap with Er, thus picking just the 5I6 level is oversimplification and not justified enough without further evidences.
4. the proposed concept indeed enables to complicate ET between sensitizers and activators to enable in-situ color changes, but the UCQY of new compounds is a few fold weaker than existing $YbEr@...$ co-doped optimized NPs (which reach a few % in best reports as compared to c.a. 0.8% here). Comparing intensities / brightness of 2 different samples is most probably not sufficiently reliable (due to various NP sizes and scattering, various amount of NP per volume), unless made on single NP level.
5. what are the reasons for non-single exponential decays (Fig.S31)
6. Are combined 2 references under single number allowed - see ref.49
7. what exactly the authors means by "up conversion enhancement" - do they mean brightness or quantum yield, or maybe other parameter ?
8. Based on extremely valuable and convincing studies done on core-shell NPs by D. Hudry et al., it seems it is not possible to claim sharp edge between core and shell parts of the UCNPs in homogeneous host, thus 'interracial' ET concept (core concept of this work) is not perfectly justified and should be discussed, considering the fact ions intermixing may occur between subsequent layers, especially high doping levels are studied.
9. The claimed 'novelty' that UC intensity enhancement is more pronounced at lower intensities is actually well known, thus cannot be considered as novelty. Moreover, the UCQY pump power dependence does not show typical saturation (e.g. Fig.S5 $ErHo@Yb@Y$ sample)- any explanation why ?

Overall, I accept the comments made in response to reviewers evaluation, but the responses were provided in the rebuttal letter, but actually not implemented in the manuscript to improve the quality of the message and the explanation of the novelty for the final readers. My current comment accounts for responses to my own comments about novelty as well as clarity and logical structure commented by Reviewer #2. In my opinion, the response provided in the rebuttal letter is not sufficient, as the manuscript itself was not really improved. Moreover, in general I sustain my previous comment about lack of in-depth insights (meaning mechanistic and quite superficial explanation are offered) and lack of strong novelty, however the rebuttal letter clarified some points, which previously were not sufficiently clear.

Reviewer #4 (Remarks to the Author):

The authors addressed the reviewer's comments and the quality of the manuscript was improved after the revision. Thus, the reviewer thinks that this manuscript is acceptable.

Point-by-Point Responses to the comments of Reviewers (#1, #3, and #4)

Manuscript Number: NCOMMS-23-16708A-Z

Title: Spatiotemporal control of upconversion photochromic evolution from a single emitter through interfacial energy transfer

(Changes in the revised manuscript as a response to the reviewers' comments are highlighted in red color.)

Reviewer #1:

The authors have adequately responded to my concerns.

Reply: Many thanks for the valuable comments and advices to our manuscript.

Reviewer #3:

Q1. *That is good the UCQY measurements were added, but more detailed information and exemplary data are requested to demonstrate how UCQY were measured and analyzed. These are not trivial measurements. How pump power dependent UCQY were obtained (by changing LD current?), especially of QY < 1% ?*

Reply: The upconversion quantum yield (QY) was measured by using a fiber optical spectrometer combined with an integrating sphere (see Fig. R1a). The test method was supplemented in the section of Characterization in SI. A side-polished quartz cuvette containing a cyclohexane solution of the nanoparticles was placed in the integrating sphere, which was excited by a 980 nm laser (Beijing Laserwave Optoelectronics Technology Co., Ltd.). The pump laser power was measured by a power meter. A tuning of the pump power was obtained by changing the laser-diode current and the power was determined by a power meter. The spectra data were collected by the fiber optical spectrometer (VIS/NIR QUEST, Ocean optics) and QY was calculated by the following equation:

$$QY = N_{em}/N_{abs} = \int P_{em}(\lambda)\lambda d\lambda / \int [P_{ref}(\lambda) - P_{sam}(\lambda)]\lambda d\lambda$$

where $P_{em}(\lambda)$ is the measured upconversion emission power spectrum of the sample, $P_{ref}(\lambda)$ and $P_{sam}(\lambda)$ are the power spectra of excitation light recorded after passing through the reference and sample, respectively. The measured power spectra are shown in Fig. R1b.

Fig. R1 (a) Schematic of the setup for QY measurement. (b) Measured excitation and emission spectral power from the $\text{NaErF}_4:\text{Ho}(0.5 \text{ mol\%})\text{NaYbF}_4@\text{NaYF}_4$ core-shell-shell sample and the ref sample under 980 nm excitation (11.6 W/cm^2).

Q2. What do the variations in XRD patterns (e.g. Fig.S9) mean from crystallographic perspective ?

Reply: In XRD patterns, the diffraction profiles and peak positions of the samples agree well with the standard card (PDF#27-0689, PDF#27-1427, PDF#16-0334), indicating that the samples are in pure hexagonal phase. The diffraction peaks of the core-shell and core-shell-shell samples become sharper than that of core only samples due to the increase of the particle size, being consistent with results of TEM images. For the XRD measurement, the nanoparticles in cyclohexane solution were added on the silicon slide with natural evaporation, and these nanoparticles were randomly aligned on the surface of silicon slide, which would lead to slight variations in XRD patterns when compared to the standard card, which do not affect the accuracy of this method. Such slight variations in XRD patterns were widely observed for nanoparticles, such as that (e.g., Fig. S5) in ref 10 (*Nano Lett.* **2021**, 21, 4838-4844).

Fig. S5 in ref 10:

Fig. S5 XRD patterns of a series of core and core-shell nanoparticles. Reproduced with permission from Hong A. R., Kyhm, J. H., Kang, G. & Jang, H. S. Orthogonal R/G/B upconversion luminescence-based full-color tunable upconversion nanophosphors for transparent displays. *Nano Lett.* **2021**, 21(11), 4838-4844. Copyright American Chemical Society (2021)

Q3. Ho^{3+} energy levels are much more complex than just 5I_6 and actually some levels exists, which overlap with Er , thus picking just the 5I_6 level is oversimplification and not justified enough without further evidences.

Reply: Thanks for the advice! The UC process strongly relies on the intermediate state. For the green and red upconversion emissions of Er^{3+} , their intermediate states are $^4I_{11/2}$ and $^4I_{13/2}$, respectively (Fig. 1d in manuscript). Thus, a tuning of the populations of Er between these two states is an effective way to alter the relative emission intensity of green and red emissions. It is found that Ho^{3+} 5I_6 level lies between the levels $^4I_{11/2}$ and $^4I_{13/2}$ of Er^{3+} . The doping of Ho^{3+} helps to populate Er^{3+} from upper $^4I_{11/2}$ to lower $^4I_{11/2}$ via the energy transfer looping (ETL) process [$Er^{3+} (^4I_{11/2}) \rightarrow Ho^{3+} (^5I_6) \rightarrow Er^{3+} (^4I_{13/2})$], resulting in an increase of red upconversion emission. To check the possible interactions between other energy levels, we measured the lifetimes of Er^{3+} at different levels before and after doping of Ho^{3+} , see the newly added Supplementary Fig. 2. It can be observed that the lifetime of $^4I_{11/2}$ shows a decrease due to the $Er^{3+} (^4I_{11/2}) \rightarrow Ho^{3+} (^5I_6)$ energy transfer and correspondingly the lifetime of $^4I_{13/2}$ shows an increase due to the $Ho^{3+} (^5I_6) \rightarrow Er^{3+} (^4I_{13/2})$ back energy transfer, while the green and red emissions do not show obvious change in lifetime. This result further confirms that the interactions between Er^{3+} and Ho^{3+} mainly occur between the three ETL relevant levels [$Er^{3+} (^4I_{11/2})$, $Er^{3+} (^4I_{13/2})$ and $Ho^{3+} (^5I_6)$]. Other energy transfer channels can be ignored.

In the revised manuscript, we updated the Fig. 1d by showing more energy levels to more clearly show the interaction between $Er^{3+} (^4I_{11/2}$, $^4I_{13/2})$ and $Ho^{3+} (^5I_6)$.

The updated Fig. 1:

Fig. 1

The newly added Supplementary Fig. 2:

Supplementary Fig. 2. (a-d) Decay curves of Er^{3+} at its ${}^4\text{I}_{13/2}$, ${}^4\text{I}_{11/2}$, ${}^4\text{S}_{3/2}$ and ${}^4\text{F}_{9/2}$ levels from $\text{NaErF}_4@/\text{NaYF}_4$ and $\text{NaErF}_4:\text{Ho}(0.5 \text{ mol}\%)/\text{NaYF}_4$ core-shell nanoparticles under pulse excitations.

Q4. *The proposed concept indeed enables to complicate ET between sensitizers and activators to enable in-situ color changes, but the UCQY of new compounds is a few fold weaker than existing $\text{YbEr}@/\dots$ co-doped optimized NPs (which reach a few % in best reports as compared to c.a. 0.8% here). Comparing intensities / brightness of 2 different samples is most probably not sufficiently reliable (due to various NP sizes and scattering, various amount of NP per volume), unless made on single NP level.*

Reply: Thanks for the useful advice! It is important to give a more reliable comparison of our sample with the typically used $\text{NaYF}_4:\text{Yb}/\text{Er}(20/2\%)/\text{NaYF}_4$ core-shell nanoparticles. In fact, in the last version we prepared these nanoparticles with nearly identical size (~40 nm; see the newly added Supplementary Fig. 22a,b) and compared their upconversion intensity under identical measurement condition (Fig. 3e). This should give reliable and accurate data for comparison albeit the measurement on the single nanoparticle level is not available in our lab. Here we further measured the QY of $\text{NaYF}_4:\text{Yb}/\text{Er}(20/2\%)/\text{NaYF}_4$ core-shell nanoparticles (Supplementary Fig. 22d), for instance it is 1.13% under pump power density of $103.96 \text{ W}/\text{cm}^2$ which is a comparable QY with literatures. Therefore, it is found that although our sample shows lower QY values but with much higher intensity due to the much high doping concentration of both sensitizer (Yb) and activator (Er), especially at lower pump power densities. In the revised version (page 8), the sentence was added "Here it is worth noting that the upconversion luminescence of this sample is much higher than that of the commonly reported $\text{NaYF}_4:\text{Yb}/\text{Er}(20/2 \text{ mol}\%)/\text{NaYF}_4$

core-shell control nanoparticles, **albeit a decrease in quantum yield** (Fig. 3e; Supplementary Figs. 22 and 23)." to point out this result.

Another important result lies in that we demonstrate that the design of $\text{NaErF}_4:\text{Ho}@ \text{NaYF}_4 @ \text{NaYbF}_4 @ \text{NaYF}_4$ has much better performance in both intensity and QY when compared to the regular $\text{NaErF}_4 @ \text{NaYF}_4$ sample, which provides an effective approach towards the bright upconversion in heavily doped nanoparticles and also is helpful to overcome the concentration quenching effect in luminescent materials.

The newly added Supplementary Fig. 22:

Supplementary Fig. 22. (a,b) TEM images of the $\text{NaErF}_4:\text{Ho}(0.5 \text{ mol}\%) @ \text{NaYF}_4 @ \text{NaYbF}_4 @ \text{NaYF}_4$ core-multishell ($\text{ErHo}@ \text{Y}@ \text{Yb}@ \text{Y}$) and $\text{NaYF}_4:\text{Yb}/\text{Er}(20/2 \text{ mol}\%) @ \text{NaYF}_4$ core-shell ($\text{Yb}_{20}\text{Er}_2 @ \text{Y}$) nanoparticles at each stage. (c) The corresponding size distributions of (a) and (b) samples. (d) Upconversion quantum yield obtained from (a) and (b) samples under 980 nm excitation with variable pump power densities.

Q5. What are the reasons for non-single exponential decays (Fig.S31)

Reply: Thanks for the valuable comment. The decay curves in our work were measured by the oscilloscope (Tektronix, MDO32) coupled with spectrometer and the data collection was recorded by the average-value mode. We found that when the laser pulse width is 0.5 ms the emission signal is very weak, and we used a high average-frequency mode to record the signal, which would result in a slight deviation from the real decay curve. Here, we re-measured the decay by reducing the average

frequency together with the high-resolution mode to acquire the more accurate data. The resulted decay shows single exponential function and was replaced in Fig. S31b and the fitted value was replaced in Fig. S31c (note that Fig. S31 was renamed as Supplementary Fig. 35 in revised SI).

The updated Supplementary Fig. 35:

Supplementary Fig. 35. (a,b) Decay curves of Er³⁺ at its (a) ⁴S_{3/2} (540 nm) and (b) ⁴F_{9/2} (654 nm) from NaErF₄:Ho(0.5 mol%)/NaYF₄:Yb(10 mol%)/NaYF₄ core-shell-shell nanoparticles under 980 nm excitation with different pulse widths (0.5-10 ms). (c) Lifetime values from (a) and (b).

Q6. Are combined 2 references under single number allowed - see ref.49

Reply: Thank you very much for your careful reading. The second ref 49 repeats with the ref 50, and we deleted it in the site of ref 49 revised manuscript.

Q7. What exactly the authors means by "up conversion enhancement" - do they mean brightness or quantum yield, or maybe other parameter ?

Reply: In our manuscript, the upconversion enhancement refers to the brightness. And in the revised manuscript, we used the "upconversion luminescence enhancement" to replace the "upconversion enhancement" for a clearer presentation.

Q8. Based on extremely valuable and convincing studies done on core-shell NPs by D. Hudry et al., it seems it is not possible to claim sharp edge between core and shell parts of the UCNPs in homogeneous host, thus 'interracial' ET concept (core concept of this work) is not perfectly justified and should be discussed, considering the fact ions intermixing may occur between subsequent layers, especially high doping levels are studied.

Reply: Many thanks the valuable comment. Ions intermixing between nearby shell layers is really an important issue for the construction of core-shell structure. We have already noted a series of valuable works, especially the ones by Dr Hudry et al suggested by this reviewer (e.g., see *Chem. Mater.* **2017**, 29, 9238-9246 and *Nat. Commun.* **2023**, 14, 4462; they were newly cited as refs 50,51). In their research, the heterogeneous core-shell structure was designed to prevent cation intermixing which tends to occur in homogeneous core-shell structure. It is worth noting that the related

studies were based on the cubic NaYF₄. In contrast, the hexagonal (β) NaYF₄ shows more stable properties in which no dissolution and phase transition would occur at the reaction temperature around 300 °C. Recently, Wang et al proposed a method to investigate this issue by using the spectral change of the Ce/Tb doped β -NaYF₄ core-shell nanoparticles and found that the dopant ions are firmly confined in their designed locations and the elemental migration would occur after heat treatment above 350 °C (*Angew. Chem. Int. Ed.* **2015**, 54, 12788-12790; newly cited as ref 52). Another work (*Chem. Mater.* **2019**, 31, 5608-5615; newly cited as ref 53) showed that large-scale (a few nanometre regions) diffusion would occur at core-shell interface after treated at high temperature over 3 hours. In our work, the NaErF₄:Ho@NaYbF₄@NaYF₄ has heterogeneous compositions, its thermal decomposition reaction time is only 30 minutes at 300 °C, which should effectively avoid cation intermixing at the core-shell interface.

We further performed the control sample to check the cation intermixing by designing a control sample. Considering that the ion intermixing mainly occur in a very narrow region at the core-shell interface (see the newly added Supplementary Fig. 6a), here we designed the NaErF₄:Ho(0.5 mol%)@NaEr_{1-x}Yb_xF₄@NaEr_xYb_{1-x}F₄@NaYbF₄@NaYF₄ ($x = 0\sim 50$ mol%) core-multishell structure by adding two new doped interlayers (i.e., NaEr_{1-x}Yb_xF₄ and NaEr_xYb_{1-x}F₄) between Er core and Yb shell layer (see Supplementary Fig. 6b). In this control sample, we assume that Er and Yb have identical diffusion rate into the opposite shell for a simple model, and the diffused content of Er and Yb were set to be 1~50 mol% to simulate the possible ion diffusion. Their upconversion emission spectra under 980 nm excitation (see Supplementary Fig. 6c,d) showed that the emission intensity starts to exhibit a slight decline when 1 mol% Er and Yb were doped into the newly growing interlayers (i.e., NaEr_{0.99}Yb_{0.01}F₄ and NaEr_{0.01}Yb_{0.99}F₄), and then a rapid decline with further increasing the doping concentrations up to 50 mol%. These results suggest that a doping content as small as 1 mol% into the opposite nearby layer would lead to a change in emission intensity. If the ion intermixing already occurred at the core-shell interface, there should be no change in the emission intensity in the control sample with a small x value (e.g., $x=1$ mol%). In contrast, we can observe a slight change in intensity. Thus, the cation diffusion can be ignored in our work.

In the revised manuscript (page 6), we added the following sentence to point out this result:

"In addition, ion diffusion at the core-shell interfacial region can be ignored (Supplementary Figs. 7 and 8) ⁵⁰⁻⁵³."

The newly added Supplementary Figs. 7 and 8:

Supplementary Fig. 7. (a) Schematic of possible ion diffusion at core-shell interfacial region in NaErF₄:Ho(0.5 mol%)@NaYbF₄@NaYF₄ core-shell-shell nanostructure. (b) Proposed NaErF₄:Ho(0.5 mol%)@NaEr_{1-x}Yb_xF₄@NaEr_xYb_{1-x}F₄@NaYbF₄@NaYF₄ core-multishell nanostructure design to investigate possible ion diffusion by artificially doping identical concentration (x mol%) of Er³⁺ and Yb³⁺ into the newly added interlayers. (c,d) Upconversion emission spectra and upconversion emission intensity of (b) samples with a tuning of doping concentrations (x = 0~50 mol%) under 980 nm excitation.

Supplementary Fig. 8. (a-f) TEM images of NaErF₄:Ho(0.5 mol%)@NaEr_{1-x}Yb_xF₄@NaEr_xYb_{1-x}F₄@NaYbF₄@NaYF₄ (x = 0~50 mol%) core-multishell nanoparticles at

each stage. (g) The corresponding size distributions of (a-f) samples.

The newly cited refs 50-53:

(50) Hudry, D., et al. Direct evidence of significant cation intermixing in upconverting core@shell nanocrystals: Toward a new crystallochemical model. *Chem. Mater.* **2017**, 29(21), 9238-9246.

(51) Arteaga Cardona, F., et al. Preventing cation intermixing enables 50% quantum yield in sub-15 nm short-wave infrared-emitting rare-earth based core-shell nanocrystals. *Nat. Commun.* **2023**, 14(1), 4462.

(52) Chen, B., et al. Establishing the structural integrity of core-shell nanoparticles against elemental migration using luminescent lanthanide probes. *Angew. Chem., Int. Ed.* **2015**, 54(43), 12788-12790.

(53) Liu, L., et al. Elemental migration in core/shell structured lanthanide doped nanoparticles. *Chem. Mater.* **2019**, 31(15), 5608-5615.

Q9. *The claimed 'novelty' that UC intensity enhancement is more pronounced at lower intensities is actually well known, thus cannot be considered as novelty. Moreover, the UCQY pump power dependence does not show typical saturation (e.g. Fig.S5 ErHo@Yb@Y sample)- any explanation why ?*

Reply: Many thanks for the useful comment. The UC intensity enhancement is more pronounced at lower intensities is not a high novelty, which is also not the main highlight of this work. Here we proposed a core-shell structure to enhance the upconversion of Er sublattice instead of regular low doping systems (e.g, 2% Er). Because there is no room to introduce sensitizer into Er lattice without reducing its content, the design of another nearby sensitizer shell becomes a possible approach to enhance the upconversion intensity. And this design can be further optimized by tuning the separation of Er and Yb layers in which both brightness and QY were enhanced. These samples also show unique temporal dependent upconversion performance resulting in an emission colour tuning with pulse excitation. On the other hand, our designed samples show much higher intensity under lower power laser excitation when compared to the regular 20%Yb,2%Er nanoparticles (despite still with a lower QY), showing great promise in some frontier fields such as bioimaging and therapy. In this point, our work is also a good example of enhancing emission brightness by using an internal way of designing sensitizer layer in a nanostructure (maybe its QY is still low) in addition to outer ways such as surface attaching dyes. In the revised manuscript (page 9), we deleted the words " in particular at the lower pump power densities " in the discussion of the " Here it is worth noting that the upconversion luminescence of this sample is much higher than that of the commonly reported NaYF₄:Yb/Er(20/2 mol%)@NaYF₄ core-shell control nanoparticles, in particular at the lower pump power densities " in last version.

Regarding the saturation of QY under higher pump power densities, the absence of obvious "saturation" trend of our samples maybe due the pump power density that is not sufficiently high for a relatively low QY. As shown in Fig. 2a in ref R1 (*Nano Lett.*

2016, 16(11), 7241-7247), higher QY tends to show a saturation trend with increasing pump power densities. We also measured the QY of the regular NaYF₄:Yb/Er(20/2 mol%)-NaYF₄ core-shell (Yb₂₀Er₂@Y) nanoparticles (see the newly added Supplementary Fig. 22d), which is higher and starts to show a saturation trend with increasing pump power densities. Although much high value over 10² W/cm² is not available in our lab, the new data in Supplementary Fig. 22d helps to support our explanation.

Supplementary Fig. 22. (d) Upconversion quantum yield obtained from (a) and (b) samples under 980 nm excitation with variable pump power densities.

Fig. 2a Power-dependent UCQY of the β -NaYF₄:Yb³⁺,Er³⁺ nanocrystals as a function of the β -NaLuF₄ shell thickness. Reproduced with permission from Fischer, S., Bronstein, N. D., Swabeck, J. K., Chan, E. M. & Alivisatos, A. P. Precise tuning of surface quenching for luminescence enhancement in core-shell lanthanide-doped nanocrystals. *Nano Lett.* 2016, 16(11), 7241-7247. Copyright American Chemical Society (2016).

Q10. Overall, I accept the comments made in response to reviewers evaluation, but the responses were provided in the rebuttal letter, but actually not implemented in the manuscript to improve the quality of the message and the explanation of the novelty for the final readers. My current comment accounts for responses to my own comments about novelty as well as clarity and logical structure commented by Reviewer #2. In my opinion, the response is not sufficient, as the manuscript itself was not really improved. Moreover, in general I sustain my previous comment about lack of

in-depth insights (meaning mechanistic and quite superficial explanation are offered) and lack of strong novelty, however the rebuttal letter clarified some points, which previously were not sufficiently clear.

Reply: Thank you for the critical comments and valuable advices. In the last revision version, we have carefully and thoroughly addressed comments of all the reviewers by providing additional evidence and analysis to support our research findings. In particular, the new measurement of quantum yield and dynamic rate calculations further confirmed the improvement of our design in enhancing the upconversion luminescence quantitatively, and presented more evidence about our explanation for the enhancement of upconversion through energy transfer rate. We also have added more detailed experimental data and adjusted the logical structure of the maintext to improve quality of our manuscript based on the reviewers' suggestions. Those responses had been added in the last version by adding new content and figures in maintext and SI (Supplementary Figs. 1-3,5,8-9,13,15-16,18-20,22,24,26,31-32, Table S5,S6 in last SI version). We greatly appreciate the specific comments provided this time by this reviewer, which has helped us to make further improvements in our manuscript. More important experimental and analytical details mentioned in the responses to the comments have been supplemented in SI (Supplementary Figs. 2,7,8,22). These newly added data and discussions indeed improved the level of our manuscript greatly.

Regarding the novelty, in this work, we demonstrate a new strategy to realize the spatiotemporal control of upconversion from a single emitter through constructing the interfacial energy transfer (IET) channel. In particular, we obtained the following novel and important results:

- (i) We reported a new sample design to greatly enhance the upconversion of Er sublattice by the sensitizing Yb shell through IET. The total upconversion shows an enhancement with one order of magnitude, and the upconversion quantum yield was also increased.
- (ii) We demonstrated a new conceptual strategy to enhance upconversion by selectively manipulating the Er-Yb sublattices interactions at core-shell interfacial region, and further found that the critical distance R_c for backward energy transfer (0.6 nm) is much smaller than that of forward energy transfer (2.8 nm), which is the first quantitative investigation on this issue for UCNPs. In addition, the ion diffusion at core-shell interface can be taken no consideration by designing the control experiments.
- (iii) We further demonstrated the green/red colour-switchable output by precise control of the rise and decay dynamics through non-steady-state excitation, which was not available in conventional Er-doped materials. Our design also shows different decay times allowing to make a colour tuning by using the time-gating technique.
- (iv) The smart spatiotemporal control of emission colours in nanoparticles permits great chances for frontier photonic applications including information storage and security, multi-level anticounterfeiting, and speed monitoring. In particular, the speed sensitivity is as high as 1.70 (km/h)^{-1} , higher than the recent reports.

Therefore, we believe that our systematic work presents a substantial breakthrough and a conceptual advance in the field of luminescent materials instead of “lack of in-depth insights” and “lack of strong novelty” assessed by this reviewer. The discovery of spatiotemporal control of activator-sensitizer interactions between sublattices rather than regular dopants provides new perspectives to improve the understanding of the ionic interactions. It broke the stereotypes roles about lanthanides as shown in previous reports which helps greatly for much more efficient and versatile manipulation of the photon upconversion, thus providing new insight in the understanding of lanthanide-based materials. In order to make a more clear clarification, we have rewritten the maintext properly, polished the language and added series of new experimental results as well as discussions during the last two revised versions of the manuscript, and hope the reviewer will perhaps find our work more suitable for publication in *Nature Communications*.

Reviewer #4:

The authors addressed the reviewer’s comments and the quality of the manuscript was improved after the revision. Thus, the reviewer thinks that this manuscript is acceptable.

Reply: Many thanks for the valuable comments and advices to our manuscript.

REVIEWER COMMENTS

Reviewer #3 (Remarks to the Author):

The authors responded to my previous comments only partially:

1. I do not understand, why the discussion with reviewers (e.g. on QY methodology, XRD analysis, energy levels of Ho³⁺ etc.) was not transferred to MS or SI - if the reviewer has hesitations about some important aspects of the work, the readers will most probably have the same hesitations. Therefore, why the response is restricted to the response letter only ?

2. About UCQY - the intensities of UC are much (hundreds of times) weaker than the scattered excitation laser line (fig R1b) and the spectral sensitivity of detection CCD was not taken into account. From the equation used to calculate QY and the provided description, it looks no correction was applied, which for me means the UCQY data are not reliable, unless some steps in the data acquisition procedure were not described in sufficient details.

3. Again about Ho³⁺ - some of the Ho³⁺ levels display energies similar to Er³⁺ - i.e. 5F₅ (Ho³⁺) - 4F_{9/2} (Er³⁺), 5S₂ (Ho³⁺) - 3S_{3/2}(Er³⁺) 5F₁(Ho³⁺) - 4F_{7/2} (Er³⁺) - but the lifetimes of respective levels of Er³⁺ were not measured with vs without Ho³⁺. Therefore the claim "This result further confirms that the interactions between Er³⁺ and Ho³⁺ mainly occur between the three ETL relevant levels [Er³⁺ (4I_{11/2}), Er³⁺ (4I_{13/2}) and Ho³⁺ (5I₆)]. Other energy transfer channels can be ignored." (sic!) is not justified. The Ho³⁺ and Er³⁺ UC emission are spectrally very similar (!), and previously measured bi-exponential lifetimes may be the explanation for this (I was asking for explanation for this, but the authors measured that again and achieved single exponents this time - how come?). I suspect, the authors see the emission from Ho and Er in the same time, but this would require more evidencing.

Point-by-Point Responses to the comments of Reviewer #3

Manuscript Number: NCOMMS-23-16708B

Title: Spatiotemporal control of upconversion photochromic evolution from a single emitter through interfacial energy transfer

(Changes in the revised manuscript and SI as a response to the reviewers' comments are highlighted in red color.)

Reviewer #3:

The authors responded to my previous comments only partially:

Q1. I do not understand, why the discussion with reviewers (e.g. on QY methodology, XRD analysis, energy levels of Ho^{3+} etc.) was not transferred to MS or SI - if the reviewer has hesitations about some important aspects of the work, the readers will most probably have the same hesitations. Therefore, why the response is restricted to the response letter only ?

Reply: Thanks for the advice. In the last revision, most of the newly supplemented data were included in the revised manuscript or SI properly, except for the scheme of QY measurement and XRD analysis. And partial of the analysis regarding the energy levels of Ho^{3+} was added. The slight deviation of XRD diffraction patterns for different nanoparticles is usually observed for nanomaterials, and was not added in the last revised version. While the absence of the QY scheme and energy levels of Ho^{3+} were our negligence; these data indeed help to remove misunderstanding for readers. In the revised version, we added the scheme of QY scheme as Supplementary Scheme 1 together with the spectral calibration in the QY measurement. We also supplemented more experimental results to check the energy transfer between the red and green emission energy levels, in order to answer the question raised by this reviewer. These data were added in the updated Supplementary Fig. 2. The XRD analysis was added in the Supplementary Fig. 4 caption.

The newly added Supplementary Scheme 1 in SI:

Supplementary Scheme 1. (a) Schematic of the setup for QY measurement. (b)

Measured excitation and emission power spectra from the NaErF₄:Ho(0.5 mol%)NaYbF₄@NaYF₄ core-shell-shell sample and the ref sample under 980 nm excitation (11.6 W/cm²).

The updated Supplementary Fig 4 caption in SI:

Supplementary Fig. 4. XRD patterns of NaErF₄:Ho(0.5 mol%)@NaYF₄:Yb(0-100 mol%)@NaYF₄ core-shell-shell nanoparticles at each stage. Core is NaErF₄:Ho(0.5 mol%). CS and CSS represent NaErF₄:Ho(0.5 mol%)@NaYF₄:Yb(0-100 mol%) core-shell and NaErF₄:Ho(0.5 mol%)@NaYF₄:Yb(0-100 mol%)@NaYF₄ core-shell-shell nanoparticles, respectively. **The deviation of the diffraction patterns of the nanoparticles were resulted from the random dispersion of them on the surface of silicon slide with different crystalline planes to X-ray.**

Other details regarding the QY and energy transfers were answered in the responses to the following comment 2 and 3.

Q2. *About UCQY - the intensities of UC are much (hundreds of times) weaker than the scattered excitation laser line (fig R1b) and the spectral sensitivity of detection CCD was not taken into account. From the equation used to calculate QY and the provided description, it looks no correction was applied, which for me means the UCQY data are not reliable, unless some steps in the data acquisition procedure were not described in sufficient details.*

Reply: Thanks for your professional question and useful advice. The spectral sensitivity of CCD is important for the emission spectra measurement. For our QY measurement, at first, the test system was calibrated by using the standard lamp (Tungsten-Halogen Lamp; Gilway 187-1), which was placed inside the center of the integrating sphere. By comparison of the measured spectrum and the known spectral radiant flux of the standard lamp, the correction factor of our setup was obtained, which was then used for the correction of the measured spectra of the sample, which results in the emission and excitation spectra in the form of power spectra with unit of μW/nm instead of the regular spectra with a.u. unit. Note that the spectral sensitivity of the CCD of the fiber optic spectrometer (QE65 Pro, Ocean optics) was corrected concurrently during this calibration procedure, because it was already included as a part of the entire setup. Therefore, the recorded spectra were the corrected power spectra, which were next used to calculate the QY by the equation (S1):

$$QY = N_{em}/N_{abs} = \int P_{em}(\lambda)\lambda d\lambda / \int [P_{ref}(\lambda) - P_{sam}(\lambda)]\lambda d\lambda \quad (S1)$$

Regarding the "intensities of UC are much (hundreds of times) weaker than the scattered excitation laser line (fig R1b)", it is due to the low QY value (0.11±0.03%), which was obtained by the equation S1.

In the revised SI, the following QY procedure was used to replace the last version for a more clear description:

" The upconversion quantum yield (QY) was measured by using a fiber optic

spectrometer combined with an integrating sphere (see Supplementary Scheme 1). At first the setup was calibrated by using the standard lamp (Tungsten-Halogen Lamp; Gilway 187-1) to obtain the correction factor, which was used to correct the measured spectra of the sample. Next, a side-polished quartz cuvette containing a cyclohexane solution of the nanoparticles was placed inside the integrating sphere coupled with the fiber optic spectrometer (QE65 Pro, Ocean optics), which collects the emission and excitation light signals under 980 nm excitation. A cyclohexane solution without the nanoparticles was also used for reference. And then the QY can be calculated by the following equation:

$$QY = N_{em}/N_{abs} = \int P_{em}(\lambda)\lambda d\lambda / \int [P_{ref}(\lambda) - P_{sam}(\lambda)]\lambda d\lambda \quad (S1)$$

where $P_{em}(\lambda)$ is the upconversion emission power spectrum of the sample, $P_{ref}(\lambda)$ and $P_{sam}(\lambda)$ are the power spectra of excitation light recorded after passing through the reference and sample, respectively. "

Q3. Again about Ho^{3+} - some of the Ho^{3+} levels display energies similar to Er^{3+} - i.e. ${}^5F_5(Ho^{3+}) - {}^4F_{9/2}(Er^{3+})$, ${}^5S_2(Ho^{3+}) - {}^4S_{3/2}(Er^{3+})$, ${}^5F_3(Ho^{3+}) - {}^4F_{7/2}(Er^{3+})$ - but the lifetimes of respective levels of Er^{3+} were not measured with vs without Ho^{3+} . Therefore the claim "This result further confirms that the interactions between Er^{3+} and Ho^{3+} mainly occur between the three ETL relevant levels [$Er^{3+}({}^4I_{11/2})$, $Er^{3+}({}^4I_{13/2})$ and $Ho^{3+}({}^5I_6)$]. Other energy transfer channels can be ignored." (sic!) is not justified. The Ho^{3+} and Er^{3+} UC emission are spectrally very similar (!), and previously measured bi-exponential lifetimes may be the explanation for this (I was asking for explanation for this, but the authors measured that again and achieved single exponents this time - how come?). I suspect, the authors see the emission from Ho and Er in the same time, but this would require more evidencing.

Reply: Many thanks for the critical comments. Indeed, the visible emission spectra of Er^{3+} and Ho^{3+} are very similar, resulted from their similar energy levels (see the updated Supplementary Fig. 1a). It is easy to occur for the energy transfer between these energy levels. In general, the donor-acceptor energy transfer leads to a decrease of the lifetime of the donor, which thus can be used to confirm the occurrence of energy transfer. This method works well in examining the energy transfer from $Er^{3+}({}^4I_{11/2})$ to $Ho^{3+}({}^5I_6)$ [ET1], and from $Ho^{3+}({}^5I_6)$ back to $Er^{3+}({}^4I_{13/2})$ [ET2], and the lifetime of Er^{3+} at its ${}^4I_{11/2}$ level shows a decrease for the sample after codoping of Ho^{3+} (0.5-20 mol%) due to ET1 (Supplementary Fig. 2f). The lifetime of ${}^4I_{13/2}$ shows an increase due to ET2 and then a decrease due to the energy transfer from $Er^{3+}({}^4I_{13/2})$ to $Ho^{3+}({}^5I_7)$, see Supplementary Fig. 2g. However, note that the green and red emission profiles of Er^{3+} and Ho^{3+} are too close to separate them clearly, see Supplementary Fig 2b,d,e. This means that when we measure the decay curves at these wavelengths, they show the combined decay curves of them instead of only one of them. Thus, in this case, the use of the lifetime change for investigating energy transfer becomes useless.

Alternatively, here we attempt to use the change of emission profiles to check the

possible energy transfer between the green and red energy levels of them. Although the spectra are similar, they still show some different characteristics in the detail of the emission profiles by comparison with the emission of Ho^{3+} from the regular $\text{NaYF}_4:\text{Yb}/\text{Ho}(20/2 \text{ mol}\%)\text{@NaYF}_4$ core-shell nanoparticles, see Supplementary Fig 2d,e. Namely, the red and green emission profiles of Ho has a slight blue-shift (e.g., at $\sim 535 \text{ nm}$ for the green emission and $\sim 640 \text{ nm}$ for the red emission). This difference can be used to check the energy transfer. Therefore, we prepared the control samples of $\text{NaErF}_4:\text{Ho}(0, 0.5, 5, 20 \text{ mol}\%)\text{@NaYF}_4$ core-shell nanoparticles (Supplementary Fig. 2a). We can observe that the upconversion intensity of the Er^{3+} green emissions show a rapid decrease, and the Er^{3+} red emission shows an initial increase before a rapid decrease with increasing the Ho^{3+} doping concentration (Supplementary Fig. 2c). Interestingly, the normalized emission profiles present more details that there nearly no change in the red emission profile of Er^{3+} when Ho^{3+} is 0.5 mol%, and with further increasing Ho^{3+} concentration its red emission gradually appear (still very weak by comparison to the Er^{3+} red emission, see Supplementary Fig. 2e). Similar result was observed for the Er^{3+} green emissions (Supplementary Fig. 2d). These observations confirmed that the energy transfer from Er^{3+} ($^4\text{F}_{9/2}$) to Ho^{3+} ($^5\text{F}_5$) and from Er^{3+} ($^2\text{H}_{11/2}, ^4\text{S}_{3/2}$) to Ho^{3+} ($^5\text{F}_4/^5\text{S}_2$) can be ignored when Ho^{3+} is as low as 0.5 mol%. And when more Ho^{3+} ions were doped, the above energy transfer processes begin to be more obviously observable, but still has no notable contribution to the redder emission of Er^{3+} . In addition, the energy transfers from more upper energy levels can take no consideration because no relevant emissions (400-500 nm) were recorded.

Therefore, for the 0.5 mol% Ho^{3+} doped sample, the decrease of the green emissions and increase of red emissions of Er^{3+} are mainly due to the ETL processes [Er^{3+} ($^4\text{I}_{11/2}$) to Ho^{3+} ($^5\text{I}_6$) and then back Er^{3+} ($^4\text{I}_{13/2}$)], which lead to the redder emission of Er^{3+} , as discussed in the manuscript. These results were added in the updated Supplementary Fig. 2 in SI, and the sentence "..., showing that energy transfers between the red and green emission energy levels has no contribution to the redder emission of the sample when Ho^{3+} dopant is as low as 0.5 mol%." was added in the caption to point out the results obtained in this revision. The full energy levels of Ho^{3+} was added in the updated Supplementary Fig. 1a in SI. And in the revised manuscript (pp 4), the sentence "..., and other energy transfers between Er^{3+} and Ho^{3+} has no contribution (Supplementary Figs. 1 and 2)⁴⁴." was added.

The updated Supplementary Figs. 1, 2 in SI:

Supplementary Fig. 1. (a) Schematic of upconversion processes and energy transfer looping (ET1+ET2) from Er^{3+} ($^4I_{11/2}$) to Ho^{3+} (5I_6) and then back to Er^{3+} ($^4I_{13/2}$) for redder emission of Er^{3+} in the proposed design of $\text{NaErF}_4:\text{Ho}(0.5 \text{ mol}\%)\text{@NaYbF}_4\text{@NaYF}_4$ core-shell-shell nanostructure upon 980 nm excitation. (b) Upconversion emission spectra of $\text{NaErF}_4:\text{Ho}(0.5 \text{ mol}\%)\text{@NaYbF}_4\text{@NaYF}_4$ (ErHo@Yb@Y) and $\text{NaErF}_4\text{@NaYbF}_4\text{@NaYF}_4$ (Er@Yb@Y) core-shell-shell nanoparticles under 980 nm excitation. (c) Time-dependent emission profiles of Er^{3+} at 654 nm from (b) samples.

Supplementary Fig. 2. (a) TEM images of NaErF₄:Ho(0,0.5,5,20 mol%) core and NaErF₄:Ho(0,0.5,5,20 mol%)@NaYF₄ core-shell nanoparticles and corresponding size distributions. (b) Upconversion emission spectra of (a) samples and the control NaYF₄:Yb/Ho(20/2 mol%)@NaYF₄ (YbHo) core-shell nanoparticles under 980 nm excitation. (c) Dependence of upconversion emission intensity on Ho³⁺ doping concentration from (b) samples. (d,e) A comparison of the normalized upconversion emission spectra of (b) samples under 980 nm excitation, showing that energy transfers between the red and green emission energy levels has no contribution to the redder emission of the sample when Ho³⁺ dopant is as low as 0.5 mol%. (f,g) Decay curves of Er³⁺ at its ⁴I_{11/2} and ⁴I_{13/2} levels from (a) samples under pulse 980 nm excitation.

As for decay profile, the previously measured bi-exponential decay curve under the 980 nm excitation with 0.5 ms width may be caused by the weak emission signal and incorrect use of acquisition mode of the oscilloscope. The emission intensity shows a rapid decrease with reducing the excitation laser pulse width as shown in the Supplementary Fig. 26 in SI. This means that for the measurement with shorter pulse width excitation, the collected spectral signals becomes much weaker (as shown in Fig. R1a,b). In this case the oscilloscope connected with CCD cannot trigger stabilize waveform and the average mode (512 times) of the oscilloscope was used to collect

the signal in the decay curve in the initial revision. However, this mode may cause some distortion in the signal curve because we used the average times of 512. As shown in Fig R1c, one can see that increasing the average times from 64 to 512 can amplify such distortion in the decay curve, making it look like bi-exponential. This should be why the "bi-exponential" decay curve was recorded. To avoid this error, we employed the high-resolution mode of the oscilloscope to record the signal by properly enlarging the slit (from 0.5 mm to 2 mm), and obtained the results as supplemented in the last revision, which shows a single exponential curve (the black line in Fig. R1c). In order to further confirm the better accurate result by the high-resolution method, we further compared the decay curves of the sample with stronger signals, namely with longer excitation pulse width of 1 ms. As shown in Fig. R1d that there nearly no difference for the results obtained under both average mode and high-resolution mode. Therefore, the "bi-exponential" decay curve in the initial version was due to the weak signal and the incorrect use of acquisition mode of the oscilloscope. We thank this reviewer for the useful comment which helps us to improve our decay curve measurement of weak signals and avoid error.

Fig. R1 (a,b) Upconversion emission spectra of NaErF₄:Ho(0.5 mol%)/NaYF₄:Yb(10 mol%)/NaYF₄ core-shell-shell nanoparticles under 980 nm excitation with pulse durations of 0.5 and 1.0 ms with slits of (a) 0.5 mm and (b) 2 mm. (c,d) Decay curves of Er³⁺ at its ⁴F_{9/2} (654 nm) from (a) sample under 980 nm excitation with different pulse widths of (c) 0.5 ms and (d) 1 ms under different acquisition modes and slits.

REVIEWERS' COMMENTS

Reviewer #3 (Remarks to the Author):

Thank you for clarifications and additional evidences. I have no further questions and suggest the manuscript is suitable for publishing in Nature Communication.